**Subject Category:**
Biology (whole organism)

behaviour/cognition/ecology

animal cognition, animal captivity, ecological validity, corvids

**Author for correspondence:**
Kelsey B. McCune
e-mail: kelseybmccune@gmail.com

# Captive jays exhibit reduced problem-solving performance compared to wild conspecifics

Kelsey B. McCune[1], Piotr Jablonski[2,3], Sang-im Lee[2,4] and Renee R. Ha[1]

[1]Psychology Department, University of Washington, Seattle, WA, USA
[2]Laboratory of Behavioral Ecology and Evolution, School of Biological Sciences, Seoul National University, Seoul, South Korea
[3]Museum and Institute of Zoology, Polish Academy of Sciences, Warsaw, Poland
[4]Daegu-Gyeongbuk Institute of Science and Technology School of Undergraduate Studies, Daegu, South Korea

KBM, 0000-0003-0951-0827

Animal cognitive abilities are frequently quantified in strictly controlled settings, with laboratory-reared subjects. Results from these studies have merit for clarifying proximate mechanisms of performance and the potential upper limits of certain cognitive abilities. Researchers often assume that performance on laboratory-based assessments accurately represents the abilities of wild conspecifics, but this is infrequently tested. In this experiment, we quantified the performance of wild and captive corvid subjects on an extractive foraging task. We found that performance was not equivalent, and wild subjects were faster at problem-solving to extract the food reward. By contrast, there was no difference in the time it took for captive and wild solvers to repeat the behaviour to get additional food rewards (learning speed). Our findings differ from the few other studies that have statistically compared wild and captive performance on assessments of problem-solving and learning. This indicates that without explicitly testing it, we cannot assume that captive animal performance on experimental tasks can be generalized to the species as a whole. To better understand the causes and consequences of a variety of animal cognitive abilities, we should measure performance in the social and physical environment in which the ability in question evolved.

## 1. Introduction

To understand the evolution of cognition, researchers over the past several decades have sought to quantify cognitive abilities in animals across taxa. Significant progress in this field has clarified the behaviours indicative of specific cognitive traits, the cognitive capacity possible in species across the animal

kingdom, and the neural, hormonal and environmental mechanisms affecting abilities [1]. To date, experiments are predominantly conducted with captive subjects rather than in the wild [2]. Cognitive assessments on animals in captivity have the advantage of being tightly controlled so that there are few alternative explanations to confound the observed results. Commonly, results from these assessments conducted in the laboratory are generalized to the abilities of the focal species as a whole [3]. However, captive animal performance on experimental assessments of specific cognitive abilities may not always be representative of the likely performance of wild conspecifics. To the best of our knowledge, very few studies have compared the performance of laboratory-based and wild, free-living conspecific subjects on tests of any cognitive ability (table 1).

Several factors can result in performance by captive animals on experimental tasks that do not generalize to wild conspecifics. Subjects that are born in captivity, or have lived the majority of their lives confined, experience unique ontogenetic social and physical environments, and learning opportunities [11–13]. Access to physical environmental enrichment during development can alter the function and structure of the brain [11–14]. Furthermore, previous experience receiving rewards while interacting with experimental tasks can change the motivation of the subjects and the goal-directed nature of motor behaviours on the apparatus [15]. These subjects approach assessments with previous experience that differs from wild counterparts [11,16] and could even be distinct from subjects in other captive conditions [17–21]. For example, Benson-Amram *et al.* [6] used a puzzle box task to directly compare captive-reared and wild hyenas on problem-solving ability. This experiment found that hyenas born in a research facility outperformed wild hyenas at the puzzle box, probably because the captive hyenas had significantly more previous experience with novel human-made objects. The different enrichment background experience of captive subjects can lead to performance at physical cognition tasks that is not possible in wild conspecifics, and is hard to relate to species-specific ecology and evolution. As such, the results from tests on one captive population in isolation may not be sufficient to represent an evolved cognitive trait of the focal species in general [20].

Previous studies have found that stress from captivity can affect the brain structure and performance of subjects on cognitive assessments, even over short time scales [22–24]. Although taking wild subjects temporarily into captivity may avoid confounds arising from distinct human-induced previous experiences [7,8], stress may affect the performance of these individuals on experimental tasks. Only one study has directly compared performance between subjects temporarily held in captivity and wild conspecifics. This study on spatial serial reversal learning found that performance did not differ between great tits in the wild or in captivity [7]. Yet, not all species, or individuals within a species, will respond equally to the disturbance required to move subjects into a novel, captive environment. To better understand when captive performance may represent the cognitive ability of the species in general, more research is needed comparing performance of subjects in each environment, on a variety of experimental tasks.

Here, we add to previous research by comparing problem-solving abilities and learning performance of Mexican jays (*Aphelocoma wollweberi*; hereafter 'jays') in the wild to jays held in captivity for less than 3 weeks on a simple, ecologically relevant, puzzle box foraging task. We evaluated support for three competing hypotheses: (i) testing subjects temporarily in captivity, with equivalent previous experiences to wild conspecifics, will result in no differences in performance in either problem-solving or learning (e.g. [10]); (ii) jays in temporary captivity, where we control food motivation and limit distractions from predators or competitors, will show increased problem-solving and learning performance relative to wild conspecifics; and (iii) the experience of capture and removal into a novel space for temporary captivity itself could negatively affect the performance of captive jays relative to wild conspecifics on our task. These results will further elucidate whether it is appropriate to generalize captive animal performance on experimental assessments of specific cognitive abilities to the species as a whole.

# 2. Methods

## 2.1. Experimental set-up

From May to September 2015, we conducted experimental trials in 7 flocks of Mexican jays around the Southwestern Research Station, near Portal, AZ. All individuals were colour-banded for individual identification, and trained to come to a whistle for food, which ensured timely participation in the task. Mexican jays live in stable flocks of 5–25 jays on a year-round territory. Within the flock, there

**Table 1.** Review of current literature comparing performance and behaviour between wild and captive animals on several types of problem-solving and learning tasks. Only two studies have statistically compared performance of animals in captivity and in the wild (italicized). Abbreviations under the 'type' column describe whether the study design involved an experimental manipulation (exp) or consisted of observations of spontaneous behaviours (obs). The 'captive duration' column describes the length of time subjects have been in captivity: since birth (bred), after birth but for most of the lifetime (life), or for less than a month (temp).

| study (citation number) | type | species | task | same task? | captive duration | results |
|---|---|---|---|---|---|---|
| Gajdon *et al.* [4]; Huber & Gajdon [5] | exp | kea | tube removal | same | life | captive > wild |
| *Benson-Amram et al.* [6] | *exp* | *spotted hyena* | *puzzle box* | *same* | *bred/life* | *captive > wild* |
| *Cauchoix et al.* [7] | *exp* | *great tit* | *reversal learning* | *same* | *temp* | *captive = wild* |
| Morand-Ferron *et al.* [8] | exp | great tit, blue tit | lever pulling | diff | temp | captive > wild |
| Chevalier-Skolnikoff & Liska [9] | obs | African elephants | tool use | NA | bred/life | captive > wild |
| Morand-Ferron *et al.* [10] | obs | Carib grackle | food dunking | NA | temp | captive > wild |

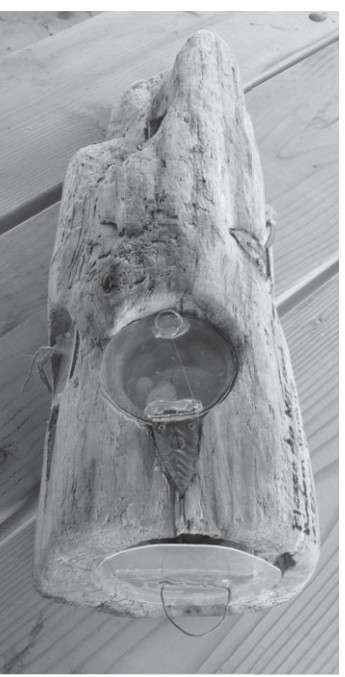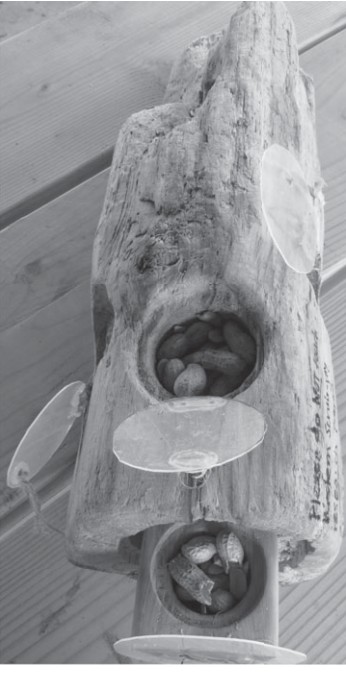

**Figure 1.** We created our puzzle box from a log by carving out four compartments and covering them with clear doors that open in different ways. The top door opens up like a hatch. The left door opens out like a car door. The front door pulls out like a drawer, and the door on the right side of the log has a hinge at the top and so pushes in (in the right photo it is pulled up for the open, habituation phase).

are overlapping generations, a dominance hierarchy and communal cooperative breeding [25]. This social system precluded identifying the sex of our subjects because Mexican jays are not sexually dimorphic. The only behavioural indicator of sex is female incubation, but only a few dominant females in each group are allowed to build nests.

We quantified individual problem-solving and learning ability with a puzzle box created out of a log, to mimic natural extractive foraging behaviours for grubs or cache-recovery behaviours [25]. We created four food-holding compartments in the log, covered by transparent doors that open in different ways (figure 1). These four types of doors could engage individuals that varied in their initial propensity to use distinct behaviours to interact with the puzzle box. Before taking any jays into captivity, on the territory of each flock we allowed all jays in the group access to the open puzzle box filled with peanuts until they habituated to this new food source. We progressed with the experimental trials only after all jays were eating from all compartments in the puzzle box without showing fear behaviours such as hesitation to contact the box and jumping quickly back after each touch.

To capture jays for the 'captive' treatment, we habituated all of the jays in five out of the seven flocks to eating peanuts from a deactivated walk-in trap. We did not capture jays from a flock until we observed their hatch year young fledge the nest and forage independently. By mid-May, several flocks had already fledged young. Of the jays from each flock subsequently bold enough to consistently and fully enter the trap, we tried to choose two adults for capture based on a relatively high position in the flock dominance hierarchy, which was necessary for another experiment [26]. We quantified dominance rank, similarly to previous research in this system [27,28], by observing agonistic interactions between individuals at a food source that could be monopolized. Subsequently, we used these data in the R package *aniDom* to assign each individual an averaged Elo-rating [29]. Preliminary analyses showed that Elo-ratings were normally distributed (Shapiro–Wilk $W = 0.98$, $p = 0.93$), and the dominance ranks of captive and wild subjects were not different ($t = 34$, $p = 0.86$), nonetheless we still included individual dominance rank in all analyses of interactions with the puzzle box.

We trapped two jays from a particular flock in the same day, and transported them immediately to large aviaries on the research station campus (figure 3 in the appendix). Aviaries were constructed of sturdy metal grating, with a natural dirt floor and ground vegetation, and averaged 6 m wide × 10 m long × 5 m high. An enclosed, connected room allowed researchers to have visual and physical access to the aviaries through a one-way window. We outfitted aviaries with a table near the window, many branches with foliage containing acorns (when in season), and a tarp attached to the ceiling and part

of the walls on the far side from the window to give jays the option for shelter from weather. Jays were fed a maintenance diet of moistened cat food, fruit chunks, mixed seed, live arthropods when possible, and peanuts. They also had constant access to a wide shallow dish of water for drinking and bathing. All captive jays were released back into their flock after a maximum of 3 weeks in captivity. These methods were approved by the University of Washington Institutional Animal Care and Use Committee (protocol no. 4064-03), the Arizona Game and Fish Department (permit no. SP697293), U.S. Fish and Wildlife (permit no. MB51894B-0) and the Southwestern Research Station.

We did not start trials until jays habituated to captivity. We assessed habituation as eating comfortably (no jumpiness) and consistently (two or more visits in 10 min) from food dishes, and subsequently the open puzzle box on the table near the window. We also considered normal caching and preening behaviours within the aviary, and foraging on the foliage with acorns as signs of habituation to captivity. All jays began exhibiting these behaviours within 48 h of their introduction into the aviaries. The day before each trial, we removed maintenance diets at sundown, and then conducted trials on captive jays the next morning before 9.00. Jays received their maintenance diet immediately after the conclusion of the trial. Wild jays were tested during the time period that their group-mates were in captivity.

To bait the puzzle box in our task, we used different food items in each treatment. Subjects in captivity and in the wild showed different preferences based on food novelty, and this is not uncommon in other previous studies [8]. Mexican jays are mast-adapted, and rely on caches of acorns and pinyon pine nuts to survive through the winter. Our experiment took place in late spring and summer when invertebrate prey, but not mast, was prevalent in the natural environment [25]. As a result, wild jay subjects were highly motivated to engage with the task for peanuts, a long-lasting cache item. In the captive condition, we first tried using peanuts to bait the puzzle box for captive subjects; however, these individuals rarely came to the table and instead recovered cached peanuts from around the aviary that they had previously received in their maintenance diet. Attendance was greatly increased when we used meal worms (Tenebrio molitor) and wax worms (Galleria mellonella) as the reward inside the puzzle box instead. While using different food rewards for each treatment may be a potential confound, it was necessary to motivate the jays in captivity to participate with the task at the same rate as the wild jays.

Mexican jays live their entire lives in social groups, and may be negatively affected by social isolation. Consequently, we needed to hold two jays from each flock in captivity at the same time. This also created a similar social environment during testing as that experienced by wild subjects because it was impossible to prevent the subjects in the wild condition from being in contact with group-mates during trials. Previous research in this population found that jays do not copy the behaviour of a group-mate on a novel task [26]. Therefore, we measured the performance of both captive jays in the aviary during the trials. In this way, jays in each treatment were similarly able to engage with the task in the presence of group-mates and neither was experiencing differential social facilitation [30]. To ensure the performance of the dyad in captivity was not affected by scrounging [27] or monopolization of the task based on dominance, we compared captive subject performance based on relative dominance rank.

## 2.2. Data collection

We conducted problem-solving trials identically for 10 captive and seven wild jays. Immediately prior to each trial attempt, we put out the puzzle box with one food item in each compartment and the doors open. If subjects came twice within 10 min and ate comfortably from the puzzle box, we replaced the food items, closed the doors and began the trial immediately. If the jays did not come back again within 10 min, the trial attempt was aborted. One wild, free-flying jay subject from each of the seven territories was chosen pseudorandomly at the start of the first trial as the very first wild jay from each flock that came to contact the puzzle box after the doors had been closed. All other jays in that flock were discouraged from coming near the set-up by blocking access to the box, or walking towards them until they moved away.

Once the trial began, we recorded whether a subject successfully opened a door and retrieved the food item (solve), the number of times it touched a specific door (attempt), the number of separate instances that a subject came within 2 m of the task (visits), the number of touches with the beak to the puzzle box anywhere other than on the doors (contacts) and the number of times each subject stood on top of the box (land). Finally, we measured the duration that the subject spent within 2 m of the puzzle box during each trial, and the latency to each solve in seconds. We manually closed the

doors and refilled any compartment that a jay successfully opened. We deemed that a subject had 'learned' the door affordances when it was able to solve a given door type three times over the course of the experiment (not necessarily three consecutive solves at a door type). In this way, we determined that the jay remembered the motor patterns necessary to open the door, rather than accidentally opening it. After three solves of a door type, that door was emptied and remained open for the rest of the trials to encourage the subject to engage with the other door types. Each trial was up to 2 h long to give each subject ample time to interact with the puzzle box. If jays lost interest and did not return within 2 m of the puzzle box for 30 min, the trial was ended early. We conducted a maximum of six trials on each subject, though some jays had fewer trials because they had already opened all doors on the puzzle box three times (for a total of 12 solves). Additionally, three jays in the captive treatment were not allowed the opportunity to solve all 12 times in six trials because their aviary partner completed all solves first, and trials were ended early to prevent the solver jay from continually opening the doors. To verify that we did not bias our results against the captive treatment by including these three individuals that did not have the opportunity for six trials, we ran two additional sets of analyses with subsets of our data. First, we compared our results from all individuals and all trials with the results from the same models when the puzzle box interactions by these three captive jay subjects were excluded. Secondly, we compared the results from our full dataset with those when the data from all jays in our experiment were truncated to just the first three trials, which every subject experienced.

## 2.3. Statistical analyses

All analyses were conducted in RStudio [31]. We used quasi-Poisson regression to account for overdispersed count data, and the glm function in the stats base package of RStudio. We modelled all interaction behaviours described above (solves, attempts, visits, contacts, lands and time within 2 m) as dependent variables, and included treatment condition and relative Elo-rating of dominance within the flock as explanatory variables. These dependent variables were summed for each subject across trials, so there was only one data point per jay. Interaction behaviours are dependent on the time that each jay spends attending to the task; therefore, in all models we included a model offset for the amount of time, in seconds, that each individual chose to spend within 2 m of the task. Time spent within 2 m of the puzzle box was a better measure of attendance to the task than trial number because during trials jays in both treatments were frequently distracted by other factors such as social interactions, foraging for other food items, or preening. Therefore, we could clearly determine that jays were attending to the task only when they chose to approach to 2 m or closer. Additionally, not all trials were the same length of time because we ended a trial early if a subject had solved all of the doors, or the subject demonstrated that it was unmotivated by not coming to the puzzle box for 30 min.

There were two exceptions to the model offset that we used. The variable 'visits' is defined as the number of times that a jay comes within 2 m of the task, and is therefore confounded with time spent within 2 m. Secondly, we were interested in whether jays in each treatment differed in the amount of time that they chose to spend within 2 m of the puzzle box. In both cases, we instead used as an offset the total trial time, in seconds, summed across all the trials that the subject received to control for the differences in opportunity to visit the experimental set-up or spend time within 2 m of the puzzle box.

Two of our hypotheses predict not only a difference in total number of solves, but also a difference in the rate of solving performance. Therefore, we used Cox proportional hazard survival analysis in the *coxme* package in RStudio [32]. Survival analysis is ideal for modelling differences between groups in time to achieve a given event (i.e. latency to solve a task or approach an apparatus; [33–38]), because these models allow inclusion of unequal samples and censored or missing data [39]. In survival analysis, explanatory variables affect the 'hazard ratio', which describes the probability that an individual will achieve the event (i.e. door opening) at a certain time point [32]. We ran two survival models to compare problem-solving and learning performance of jays in the two treatment conditions. The dependent variable for the first was the time in seconds, summed across trials, until the subject solved each different door type for the first time (problem-solving ability). The second model only included jays that had already solved a door type once, and investigated the time in seconds, summed across trials, that it took for the jay to solve that door two more times (learning speed). In these survival models, there were up to four responses for each subject (one per door type), so we included bird ID as a random effect. We included treatment condition and dominance rank as the explanatory variables. All data and code used in these analyses are available on Mendeley datasets (http://dx.doi.org/10.17632/4hczg93xkp.2).

**Table 2.** Results from the quasi-Poisson analyses of the effect of treatment on puzzle box interaction behaviours. The treatment effect represents the performance of wild jays relative to captive jays.

| response variable | fixed effects | estimate | standard error | $t$-value | $p$-value |
|---|---|---|---|---|---|
| solves | treatment | 1.26 | 0.46 | 2.77 | *0.02* |
| | dominance | 0.06 | 0.12 | 0.52 | 0.61 |
| attempts | treatment | 1.31 | 0.33 | 3.94 | *0.001* |
| | dominance | 0.12 | 0.09 | 1.34 | 0.20 |
| contacts | treatment | 0.89 | 0.46 | 1.96 | 0.07 |
| | dominance | −0.40 | 0.23 | −1.77 | 0.10 |
| lands | treatment | 0.38 | 0.35 | 1.01 | 0.29 |
| | dominance | 0.14 | 0.09 | 1.62 | 0.13 |
| visits | treatment | −0.45 | 0.62 | −0.72 | 0.49 |
| | dominance | −0.11 | 0.14 | −0.78 | 0.45 |
| seconds within 2 m | treatment | 0.32 | 0.52 | 0.62 | 0.55 |
| | dominance | −0.24 | 0.14 | −1.69 | 0.11 |

## 3. Results

With only one exception, jays that solved any door type at least once were able to successfully solve that door two more times during the six trials. Over half (4/7) of the wild subjects solved all of the doors; three subjects did so in the span of one trial. By contrast, only three out of the 10 captive jays solved all of the doors and none did so in fewer than three trials.

Wild jay subjects undergoing trials in their natural environment interacted more with the puzzle box. They made more attempts per time spent within 2 m of the puzzle box ($\beta = 1.31$, $p = 0.001$; raw mean ± s.e.: wild = 181.7 ± 53, captive = 110.1 ± 28.8), and had significantly more successes per time spent within 2 m of the puzzle box ($ß = 1.26$, $p = 0.02$; raw mean ± s.e.: wild = 8.4 ± 1.97, captive = 5.5 ± 1.77) than captive jays. However, there was otherwise no difference in the interaction behaviours of the two treatment groups. Wild and captive subjects did not differ in the proportion of time they spent within 2 m attending to the task ($\beta = 0.32$, $p = 0.55$), the number of contacts per time to the log anywhere other than on a door ($ß = 0.89$, $p = 0.07$), the number of lands per time on top of the log ($ß = 0.38$, $p = 0.29$) or the number of visits to within 2 m of the experimental set-up per time ($ß = -0.45$, $p = 0.49$). There was also no effect of dominance rank on any of our interaction variables (table 2).

We looked at the effect of dominance on these dependent variables for only captive subjects to determine if social competition within the aviary affected which jay interacted with the puzzle box. We found no relationship between dominance rank of captive jays and successes, attempts, contacts, lands, visits per time or the proportion of time spent within 2 m. These results are presented in the appendix to the main text (table 4).

Finally, our results did not change when we subset data to exclude the three captive jays that did not have the opportunity to interact with the puzzle box over six trials. We also saw no difference in our results when we truncated all data to include only interactions during first three trials of each subject (table 4).

The trend towards wild jays outperforming jays in captivity is also evident in the results from the Cox proportional hazard models (figure 2). The rate at which individuals solved a door for the first time was significantly faster in wild jays than captive jays ($p = 0.03$; 'problem-solving ability'). Conversely, for jays that were able to solve a door at least once, there was no difference between treatment groups in the rate at which they solved a door for the third and final time ($p = 0.34$; 'learning speed'). In our survival models, we found no effect of dominance on the rate of initial problem-solving performance, or the learning speed (table 3).

## 4. Discussion

Despite the popularity and ease of using captive animal subjects for assessments that aim to test animal cognition, very little research has tested whether results from captive subjects are generalizable to the

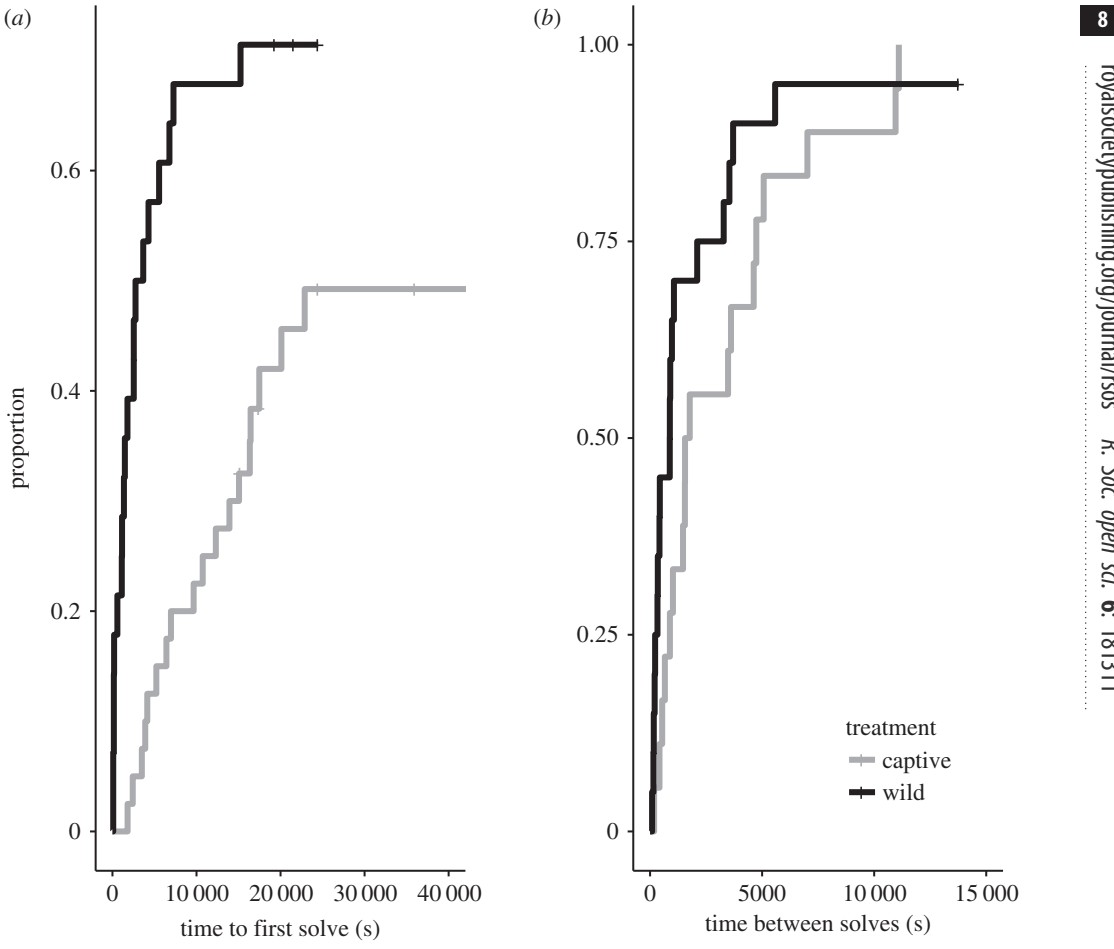

**Figure 2.** Jays in each treatment differed in the rate of initial problem-solving (*a*), but not in learning speed (*b*). The survival plots show the proportion of subjects in each treatment at each time step that solved a door for the first time, and the time it took each subject between the first and last solves on a given door type. Captive jay performance is indicated in grey, wild jay performance is in black.

**Table 3.** The output from the two Cox proportional hazard models. The hazard ratio refers to the probability of opening a door at a given time point in the wild condition relative to the captive condition. Both models include a random effect of jay ID.

| response variable | fixed effects | hazard ratio | CI | *z*-value | *p*-value |
|---|---|---|---|---|---|
| first solve | treatment | 7.08 | 1.22 – 41.4 | 2.17 | *0.03* |
| | dominance | 0.71 | 0.43 – 1.15 | − 1.38 | 0.17 |
| seconds between | treatment | 1.91 | 0.64 – 5.74 | 1.16 | 0.24 |
| | dominance | 0.94 | 0.69 – 1.25 | − 0.44 | 0.66 |

species as a whole. Here, we show that captive and wild animal problem-solving performance is not the same, even though subjects appear to be equally engaged with the task. By contrast, jays from the two treatment conditions showed no difference in learning speed.

Our study adds to a few others that have compared, directly or indirectly, the performance of wild and captive subjects on different types of tasks (table 1). Interestingly, there seems to be no consistent pattern of performance by captive and wild subjects, potentially because numerous factors could be influencing performance on each task other than the treatment condition. First, performance may vary as a function of ontogenetic experience when the amount and quality of enrichment that captive animals receive is not equal across laboratories [6,12,40]. Additionally, inconsistencies in performance can arise if captive subjects are food deprived before testing in some studies but not others, if there is

variation in the salience of a given cue, or if individuals differ in their motivation to attend to the task [41]. Finally, researchers should be aware of bias in performance arising from different personality types. For example, consistent individual differences in catchability, boldness towards novel experimental set-ups and the ability to thrive in stressful captive environments can lead to variation in performance across treatments, environments and different kinds of tests [42,43]. To make results more comparable across taxa, and to better understand how captivity affects performance on experimental assessments of cognition, researchers should try to control for these additional factors in the experimental design [41].

The increased problem-solving performance of wild jays that we observed, relative to jays in captivity, is unlikely to be attributable to differences in motivation. Captive subjects attended to the task equivalently to wild subjects because there was no difference in the proportion of a trial that jays spent within 2 m of the puzzle box, the number of visits to the set-up per total trial time, or the number of contacts and lands per time within 2 m of the task. The equal attendance of captive jays was not due to space constraints within the large, rectangular aviaries. The window to the enclosed room and the table for the puzzle box were set against the wall of one of the short sides of the rectangle (figure 3 in the appendix). As such, the captive jays had plenty of space (at least 8 m horizontally) to avoid coming within 2 m of the puzzle box, and outside of trials subjects frequently sat at the other end of the aviary where the foliage and branches were most dense. Consequently, captive subjects chose to come within 2 m of the puzzle box to get the food reward, indicating that they were motivated to engage with the task.

Similarly, because captive and wild subjects showed no difference in most interaction behaviours, it is unlikely that the dissimilar food rewards between treatments explain the depressed performance of jays in captivity. While it is not ideal to use two different kinds of rewards for each treatment condition, we do not believe that this explains our results for two reasons. First, had we used peanuts for captive birds, the observed differences would probably have been more extreme given that these individuals were unwilling to engage in the task for peanuts. Second, we would expect that live worms in the puzzle box of the captive subjects would have increased the rate of interaction with the task because mobile insects must be attacked more quickly than inanimate forage like mast [44]. Thus, using different food sources for the different treatment groups actually maximized potential performance of captive birds. Furthermore, captive jays were food deprived prior to trials, while wild birds were not, which should also bias results in favour of captive performance. These factors all result in a conservative test that minimizes the likelihood of finding reduced performance of captive jays.

One explanation for the poorer problem-solving performance of captive jays is that these individuals may have been trying to remain vigilant to possible danger in the new environment. Previous studies have found that vigilance behaviours increase as group size decreases [45–47]. We did not quantify vigilance behaviours, but it is possible that jay subjects in groups of two in captivity increased vigilance behaviours like visual scanning when near the food source (puzzle box), compared to wild conspecifics that could be near up to nine vigilant group-mates. Increased scanning of the environment could lead to the interaction behaviours that we observed: wild jays made more attempts (the functional behaviour necessary for door opening), but jays in each treatment were similar on the number of contacts, visits, lands and time spent within 2 m of the puzzle box (the non-functional behaviours). Initial problem-solving performance could also have been impaired if the captive subject's attention was divided between vigilance and attempting on the puzzle box, impeding the perceptual feedback loop between motor behaviours and the changes in the state of the door during attempts [48]. Conversely, it is notable that wild jays performed so well given the possibility of environmental and social distractions, and our lack of ability to control food motivation in the wild. Our study was not designed to test these mechanisms determining performance, but future research could explicitly measure vigilance behaviours and vary group sizes to elucidate whether this could explain depressed performance of captive subjects. Alternatively, it would be valuable to replicate this experiment with a relatively asocial species.

There was distinct individual variation in problem-solving performance by subjects in both treatments. Inspection of our survival plots indicates that if a jay has not solved a door by the third trial (20 000 s), it is unlikely it will ever be a solver. Although the doors were on the same experimental apparatus and so not distinctly separate tasks, we assessed whether jays show statistically repeatable problem-solving performance across door types [49] to better understand this variation among individuals. Details on this analysis are in the appendix to the main text. We found that captive and wild jay problem-solving performance across door types was highly repeatable ($R = 0.42 \pm 0.18$, $p = 0.002$), indicating that jays may show consistent differences in individual problem-solving ability, where some jays were 'solvers' and others were not. This was not due to learning a

generalizable rule after solving one or two door types about interaction behaviours that lead to success because subjects did not subsequently solve additional door types more quickly ($\beta = -0.47$, $p = 0.09$; see appendix).

Consistency in problem-solving performance across door types could result from several individual characteristics. First, previous research in this population has found short-term consistency in producer-scrounger roles [27], where dominant jays steal from subordinate jays that have flipped over a board to reveal food. In our study, we found no effect of dominance on any of the interaction behaviours that we measured (table 2). Therefore, it is unlikely that dominance rank affected which jays interacted with the puzzle box, or determined which jays were more successful problem-solvers. When we looked only at solving performance of captive individuals, we also found no significant effect of dominance rank. This substantiates that social competition within the aviary probably did not lead to artificially decreased performance by jays in the captive treatment.

Another explanation for the variation in solving performance is that bolder jays are most interactive with the puzzle box [50]. Previous research in this population did find consistent individual differences in boldness towards a novel object [51], but boldness scores of our subjects were not significantly related to success on this task ($\beta = 0.02$, $p = 0.37$; see appendix). This lack of relationship is perhaps not surprising given that our sampling design was biased towards the jays in each flock that were relatively more bold [43]. It was only possible to capture subjects for captivity that were bold enough to repeatedly go into the trap. Furthermore, our wild subject from each flock was the jay that was bold enough to first approach the puzzle box again after the doors were closed. As such, it is unlikely that the 'non-solvers' in our subject pool were too shy to interact with the task.

As our sample size was relatively small, it could be possible that the significant difference in problem-solving is attributable to a few exceptional individuals that happened to end up in the same treatment group. To determine if this occurred, we conducted a randomization test where we decoupled the observed treatment condition from bird ID and problem-solving performance (see appendix). If only a few individuals are driving the significant difference in performance that we observed, then we would expect a high proportion of randomized datasets to result in these individuals being part of the same treatment groups and yielding a significant treatment effect. By contrast, in 1000 randomized datasets, only 7% produced a significant difference in problem-solving attributable to treatment. Therefore, the observed difference between groups is robust with a Type I error (i.e. falsely rejecting the null hypothesis of no treatment effect) probability of only 0.07.

In contrast to problem-solving performance, we found no difference between treatment groups in learning performance. Wild and captive subjects that were able to solve a door for the first time showed no difference in the time it took to solve that door for a third and final time, indicating similar learning speeds. Moreover, learning speed was not repeatable within individuals across door types ($R = 0.15 \pm 0.16$, $p = 0.34$; see appendix). Once a subject opened a door for the first time, there were no consistent individual differences across door types in the amount of time it took to remember how to open that door type two more times. Learning speed was also not related to boldness ($\beta = 0.001$, $p = 0.39$; see appendix) or dominance (table 3). Each door type could be opened with only one simple behaviour. As a result, our task was probably not difficult enough to reveal variation in learning speed among jays or between treatments. These preliminary results merit further investigation by assessing performance of individuals on multiple distinct problem-solving and learning tasks.

To better understand the evolution of cognitive abilities in different species, we should increase the ecological validity of experimental designs by assaying performance of wild subjects on experimental tasks. Our results on subjects temporarily held in captivity, combined with previous research on animals that have lived the majority of their lives under human care, show that researchers should be cautious about generalizing captive animal performance to the species as a whole. In both of these situations, researchers should take care to address the developmental or situational confounds that could explain performance but that may not occur in the natural environment [9,52]. Finally, the results presented here, and previous research in other systems [7,53–55], demonstrate that cognitive assessments in wild subjects are possible and important for advancing the field.

Ethics. Before data collection began, the methods reported here were approved by the University of Washington Institutional Animal Care and Use Committee (protocol no. 4064-03). We obtained permission to carry out fieldwork from the Arizona Game and Fish Department (permit no. SP697293), US Fish and Wildlife Service (permit no. MB51894B-0) and the Southwestern Research Station. This study did not use human subjects.
Data accessibility. All data and code used in these analyses are available on Mendeley Data (http://dx.doi.org/10.17632/4hczg93xkp.2).

Authors' contributions. K.M. conceived of the research question and methods, conducted data collection and data analyses and wrote the manuscript. P.J. and S.L. equally contributed to data collection on dominance interactions and colour-banding of jays. P.J., S.L. and R.H. commented on study design and manuscript preparation. R.H. assisted in securing funding. All authors approve of final manuscript for submission.

Competing interests. The authors declare no competing interests.

Funding. Funding for this research was provided by the University of Washington Psychology Department, the National Science Foundation Graduate Research Fellowship Program and the grant 'Narodowe Centrum Nauki (N304 138440)'.

Acknowledgements. The authors thank Jonathon Valente and Corina Logan for the assistance and feedback throughout this research. They also thank the University of Washington Psychology Department, the director and staff of the American Museum of Natural History's Southwestern Research Station, the director of the Museum and Institute of Zoology PAS and Prof. Wieslaw Bogdanowicz for logistical support. They thank their many student assistants in the field, especially Wonyoung Lee, Choongwon Jeong, Carly Batist and Brice Lawley for their help on data collection beyond colour banding.

# Appendix A

## A.1. Randomizing subjects between treatment groups to test validity of the observed results

As our sample size is small, we wanted to verify that our results did not arise because a few exceptionally good (or exceptionally poor) individuals were allocated together to a given treatment by chance. If this is occurring, then we would predict that, in simulated datasets where we randomize the allocation of individuals to treatment condition, the exceptional individuals would end up allocated to the same group with a high probability. This would lead to many of the simulated datasets producing significant treatment effects. We tested this by taking our observed data frame containing one row per jay and randomly shuffling the 'Captive' or 'Wild' assignment in the treatment column. We did this repeatedly to create 1000 simulated datasets, and then ran our same quasi-Poisson model where we analysed for an effect of treatment on the number of successes per time, including dominance rank as a covariate. We found that in only 7% of the datasets did the model result in a significant treatment effect. That is, our estimated Type I error rate (probability of finding a significant treatment effect by chance when none exists) is 0.07. This indicates that our results are statistically valid and unlikely to be a function of a few exceptional individuals being coincidentally grouped in the same treatment.

## A.2. Effect of boldness on problem-solving and learning performance

As part of another experiment, we collected data on boldness of individuals in this population of jays. We quantified boldness using closest approach to a novel object surrounded by peanuts [51]. Peanuts varied in distance from the novel object, so bolder jays took peanuts from closer to the object during trials. We found novel object boldness to be repeatable after 4–11 weeks ($R = 0.49 \pm 0.21$, $p = 0.02$), indicating it is a valid measure of a personality trait.

We categorized jays as 'solvers' if they were able to open at least one door three times. We analysed the probability of being a solver as a function of treatment and closest approach boldness score using logistic regression. The probability of being a solver was not significantly related to boldness score ($\beta = 0.02$, $p = 0.37$). Similarly, we assessed whether the average time that individuals took between a first and last solve at a given door (learning speed) was related to boldness. We log-transformed the average time in between solves for each jay to improve normality ($W = 0.89$, $p = 0.16$), and included explanatory variables for treatment condition and dominance rank in a linear model. Individual learning speed was not significantly related to boldness score ($\beta = 0.001$, $p = 0.39$).

## A.3. Repeatability of problem-solving and learning across door types

We used a logistic mixed-effect model in the *rptR* package [56] in R to partition variance for an estimate of individual repeatability of problem-solving performance. This function also conducts parametric bootstrapping to create confidence intervals around each estimate of repeatability, and data permutations to calculate asymptotic *p*-values. We ran our model with 1000 bootstraps and 500 permutations. Our response variable was 1 if jays solved a door type three times, and 0 if jays did not reach this criterion for a given door. We included treatment condition as the explanatory variable and jay ID as a random effect. We also included a covariate of trial number in the analysis of problem-

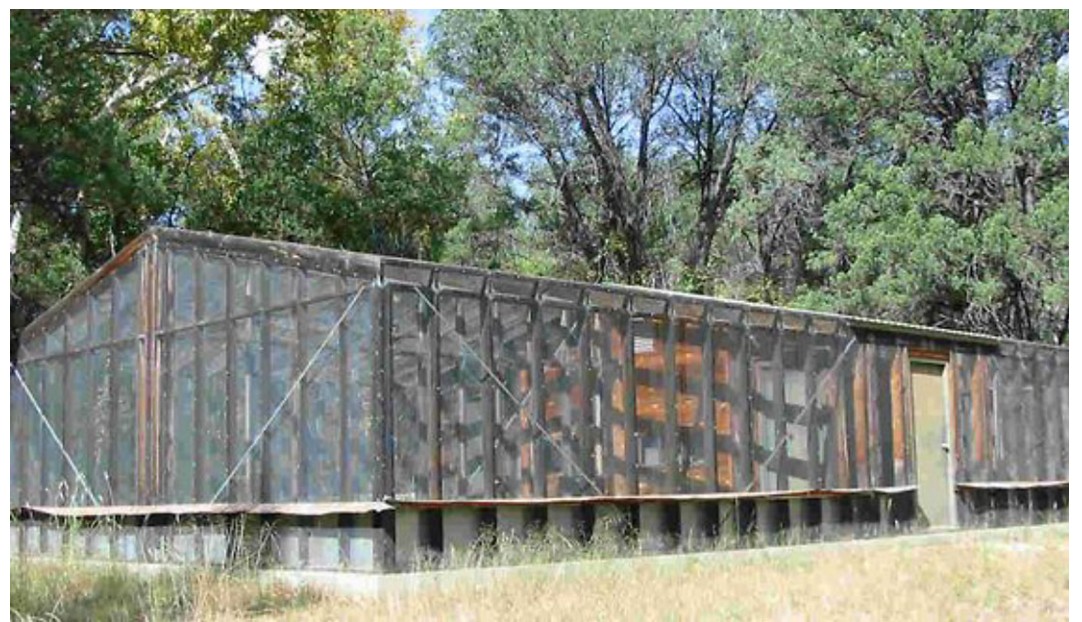

**Figure 3.** Animal behaviour observatories (aviaries) on the Southwestern Research Station campus where we temporarily held jays captive.

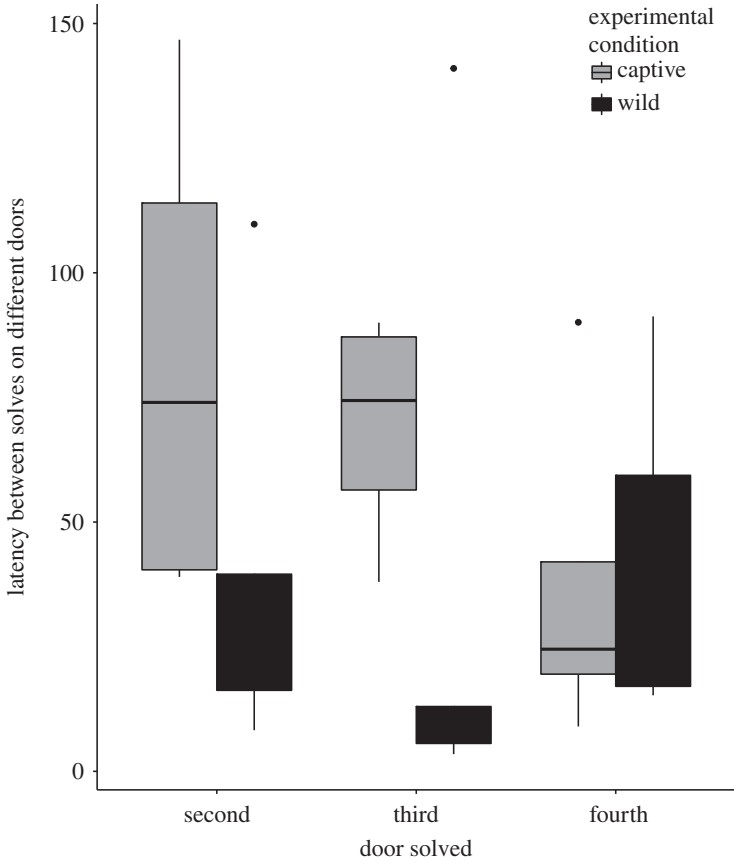

**Figure 4.** Boxplot showing the time between solves on different door types (i.e. the 'second' door solved represents the latency from the last opening of the first door type to the first opening of the second door type, etc.).

solving repeatability because jays may improve at door opening over time [57]. However, we did not include trial number in the analysis of learning ability because we measured learning as the time in between the first and last solves on each door type, so change over time was already accounted for.

**Table 4.** Results from the quasi-Poisson models with subset data. There was only one variable where the significance level changed in the subset data. When we excluded trials 4–6, wild jays had significantly more contacts to the puzzle box. However, the direction of this effect remained the same with only a minor change in value.

| data subset type | response variable | fixed effects | estimate | standard error | t-value | p-value |
|---|---|---|---|---|---|---|
| excluded 3 jays | successes | treatment | 1.10 | 0.46 | 2.37 | 0.04 |
| | | dominance | 0.08 | 0.12 | 0.66 | 0.53 |
| | attempts | treatment | 1.15 | 0.34 | 3.40 | 0.006 |
| | | dominance | 0.13 | 0.09 | 1.51 | 0.16 |
| | contacts | treatment | 0.71 | 0.44 | 1.61 | 0.14 |
| | | dominance | −0.10 | 0.13 | −0.75 | 0.47 |
| | lands | treatment | 0.52 | 0.38 | 1.37 | 0.20 |
| | | dominance | 0.10 | 0.10 | 0.99 | 0.35 |
| | visits | treatment | −0.20 | 0.64 | −0.32 | 0.76 |
| | | dominance | −0.15 | 0.16 | −0.90 | 0.39 |
| | seconds within 2 m | treatment | 0.35 | 0.62 | 0.58 | 0.58 |
| | | dominance | −0.25 | 0.18 | −1.39 | 0.19 |
| excluded trials 4–6 | successes | treatment | 1.24 | 0.41 | 3.03 | 0.01 |
| | | dominance | 0.06 | 0.11 | 0.55 | 0.59 |
| | attempts | treatment | 1.37 | 0.31 | 4.34 | 0.001 |
| | | dominance | 0.13 | 0.08 | 1.58 | 0.14 |
| | contacts | treatment | 1.39 | 0.58 | 2.40 | *0.03* |
| | | dominance | −0.04 | 0.16 | −0.26 | 0.80 |
| | lands | treatment | 0.45 | 0.34 | 1.34 | 0.20 |
| | | dominance | 0.15 | 0.08 | 1.79 | 0.10 |
| | visits | treatment | −0.51 | 0.83 | −0.61 | 0.55 |
| | | dominance | 0.004 | 0.18 | 0.02 | 0.98 |
| | seconds within 2 m | treatment | 0.35 | 0.61 | 0.57 | 0.58 |
| | | dominance | −0.18 | 0.17 | −1.07 | 0.30 |
| only captive jays | successes | dominance | 0.25 | 0.96 | 1.32 | 0.23 |
| | attempts | dominance | 0.14 | 0.13 | 1.07 | 0.32 |
| | contacts | dominance | −0.06 | 0.20 | −0.28 | 0.79 |
| | lands | dominance | 0.19 | 0.12 | 1.59 | 0.15 |
| | visits | dominance | −0.06 | 0.16 | −0.37 | 0.72 |
| | seconds within 2 m | dominance | −0.15 | 0.15 | −1.01 | 0.34 |

Jays in captive and wild treatments showed significant individual differences in problem-solving ability across door types ($R = 0.42 \pm 0.18$, $p = 0.002$). By contrast, learning speed was not repeatable ($R = 0.15 \pm 0.16$, $p = 0.34$). We also investigated whether jays showed consistent individual differences in the number of attempts per time on each door across trials. We used the same *rpt* function in R, and Poisson regression including a time within 2 m offset, to model attempts as a function of treatment and a random effect of bird ID. However, this model would not converge. We instead used a likelihood ratio test to examine the amount of additional deviance explained by the bird ID random effect over a model without this grouping variable. We found that the random effect significantly improves the model ($\chi^2 = 800.7$, $p < 0.001$), indicating attempts across trials are significantly clumped within subject. We believe this result merits further investigation with more detailed data on the

number of attempts before each successful opening of a door, rather than the more broad-scale data on attempts per trial that we present here.

## A.4. Improvement in performance after successfully solving more than one door type

One potential reason for the repeatability of problem-solving performance is that, after a first solve, jays may generally learn how to successfully interact at the puzzle box to get food from any door type. We investigated this possibility by asking whether there was a significant decrease in the time between the last solve on one door type and the first solve on the next door type. We quantified this latency for jays that solved more than one door and modelled it as a function of door solve order (second, third or fourth door type solved), treatment and a door solve order by treatment interaction term. There were multiple responses per individual, so we included bird ID as a random effect. Our latency variable was normal when log-transformed ($W = 0.95$, $p = 0.26$), so we used a linear mixed-effect model. Although there appeared to be a trend where captive subjects decreased the time between solves on different door types (figure 4), the interaction term was not significant ($\beta = 0.70$, $p = 0.08$), and door solve order was also not significant ($\beta = -0.47$, $p = 0.09$). We did find a significant treatment effect ($\beta = -2.17$, $p = 0.04$), which supports the results from our Cox proportional hazard survival models that wild subjects had faster rate of problem-solving performance than captive subjects (table 4).

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
