## [Reviewer comments · Royal Society Open Science]

Review History

RSOS-181311.R0 (Original submission)

Review form: Reviewer 1

Is the manuscript scientifically sound in its present form?

Yes

Are the interpretations and conclusions justified by the results?

No

Is the language acceptable?

No

Is it clear how to access all supporting data?

Yes

Do you have any ethical concerns with this paper?

No

Have you any concerns about statistical analyses in this paper?

Yes

Recommendation?

Major revision is needed (please make suggestions in comments)

Comments to the Author(s)

This is a paper based on a rare dataset on free-ranging individuals being tested on the same cognitive task than wild-caught individuals from the same population. The collection of such synchronized datasets is difficult to achieve, but important if we ought to evaluate the ecological validity of wild vs lab measurements of behaviour or cognition. The topic of the paper is timely and likely to appeal to a large readership.

I have reviewed a previous version of this paper for another journal and found it much improved and clarified. However there are still some issues with statistical analyses, some confusion between concepts, and there is a need to discuss some important limitations.

1-Stats: (a) to deal with unequal number of trials per individual, the authors have opted to repeat all analyses with an additional set of models (3 first trials only). Why not control for 'total number of trials' or 'trial number' as a fixed term in the model instead? The authors included 'Trials' in the analysis of 'time within 2m' and thus should apply this rule consistently. If there is a valid reason not to do so, this rationale should be explained.

(b) Repeatability: trial number must be included as a fixed term here: Biro, P. A., & Stamps, J. A. (2015). Using repeatability to study physiological and behavioural traits: ignore time-related change at your peril. *Animal Behaviour*, 105, 223-230.

2-The authors correctly define the first solution on a given door as problem-solving, and the change in performance in repeated opening of a given door as learning (L252-4). However, the introduction and abstract only concerns 'cognition' in general or learning. The authors should align the text to the data collected and appropriately refer to problem-solving when this is what is referred to or discussed. I make some detailed comments about this below.

Moreover the authors should be consistent and analyze the repeatability of learning speed in addition to that of problem-solving, and discuss the n.s. set of results on learning as well, not just the significant result on problem-solving. Together the picture is a bit more mixed concerning the 3 initial hypotheses than what the authors currently indicate.

3-The very small sample size needs to be part of the discussion; maybe there could be ways to reassure the reader that the significant effect of treatment on problem-solving success is not due to 1-2 individuals and is a general phenomenon?

Detailed comments

L221: "repeatability of learning performance to determine if 221 individual variation in performance represented inherent individual problem solving ability". This is an example where problem-solving is conflated with learning, rephrase to: "repeatability of door-opening success to determine"...

L238: why not control for total nb of trials each jay had, instead of having to conduct several analyses (l230-3), and as done on l243 for motivation? Same for persistence l245.

Definitely need to control for trial number in the survival analysis, as performance is expected to change over trials.

Same and more critical for repeatability (see main comment 1). This is very important as the

unequal number of trials could have inflated the number of 0s for birds that had fewer trials, and increased the number of 1s for birds that engaged in several trials. This would reduce within-individual variation and inflate R, potentially explaining such high $R=0.86$.

L268: I'd give results without interpreting here, cut. Furthermore, the 3rd hypothesis is not fully supported as performance differed only for problem-solving, learning was n.s. Some would say that as cognitive processes are not well understood, only the learning metric is relevant and thus your results rather support the no difference hypothesis.

Where is R for learning? Add to be consistent with problem-solving.

L312-27: this is repetitive of the introduction, I'd keep only 1-2 sentences and instead bring in the main results in this paragraph to highlight them before they get discussed in details.

L329: the wording "simple task" is uninformative and should be avoided throughout. Replace with e.g. extractive foraging task, or problem-solving task.

L335-40: while I appreciate the authors considering this alternative explanation for the n.s. result on time spent within 2m, to me the convincing point would be that they entered this space mainly when the problem was set (i.e. it is not simply their preferred spot, despite having ample other space), and/or they were actively engaged in the task when they did so.

L370: 'however' should be replaced with 'moreover'

L373: you are discussing problem-solving here, not learning.

Motivation as an alternative explanation: Difference in food reward type or possession of cached food in captivity could create differences in motivation? I'd include this point in the discussion on this issue.

L386: before going to concluding paragraph, you need to discuss learning specifically as you only discussed problem-solving: why was there a treatment effect on initial solution speed but not on improvement on the solution? Also, for consistency you should analyze repeatability of learning, which would be discussed here as well.

L438: include R estimate and CIs

L443: the MS is also about learning; what is its relationship to boldness?

Table 1. Many of these studies have NOT measured learning. Adjust caption. Also this portion of the footnote doesn't seem relevant "Previously captive birds with 1-lever experience were no more likely to solve the 2-lever task once released than naïve wild"

Review form: Reviewer 2

Is the manuscript scientifically sound in its present form?

No

Are the interpretations and conclusions justified by the results?

No

Is the language acceptable?

No

Is it clear how to access all supporting data?

Yes

Do you have any ethical concerns with this paper?

No

Have you any concerns about statistical analyses in this paper?

No

Recommendation?

Major revision is needed (please make suggestions in comments)

Comments to the Author(s)

Overall

I am fully supportive of the aims of this study, and agree with the authors that there is too often an assumption that performance in the laboratory is equivalent to performance in the wild, with only a handful of studies actually testing this. The experiment in this study is well designed, with a few issues I will detail below, and the authors have very sensibly collected data that allows us to understand how differences in motivation can account for differences in performance. I do, however, have a couple of concerns and suggestions, which I hope the authors will consider.

Major concerns

1. There is a major confound to the captivity/wild treatment the authors are testing. The birds in captivity and in the wild were rewarded with different types of food. The authors have good reasons for this decision, but do not discuss the effect this would have on performance. Possible effects include differences in performance or even differences in what is learned. Sulikowski et al. (e.g. Sulikowski, D. & Burke, D. 2010 When a place is not a place: Encoding of spatial information is dependent on reward type. *Behaviour*, 147, 1461-1479), for example, have shown that mynahs seem to learn different types of information or to use different strategies when rewarded with insects versus nectar. I would like to see the authors demonstrate that the differences they observe are due to captivity not differences in food. This could involve comparing the "motivation" of birds with different rewards, although this could be difficult as this is, as far as I can see, perfectly correlated with captivity. If the authors have any data which can tell between these two hypotheses (food vs captivity) I would like to see it included.
2. A second possible confound is competition between birds. In the wild a single bird was allowed access to the apparatus and the other birds were "discouraged" from participating (line 207). In captivity two birds interacted with the task simultaneously. It is difficult to predict a priori what effect this difference could have - anywhere from social facilitation to competitive exclusion - but I think this deserves some discussion, including why this difference is less important than the difference between captive and wild birds.
3. The authors say that their results cannot be explained by higher motivation for the task in the wild jays (lines 331-2) as there was no difference in the time spent within 2m of the box. There were, however, differences in the number of attempts, with wild jays making more attempts than captive jays. This was discussed in terms of vigilance (quite sensibly in my opinion), but could also reflect differences in motivation or neophobia for the novel apparatus (heightened in unfamiliar surroundings). If that was the case, the differences in solves could be explained in

terms of differences in attempts. I would like to see a more detailed analysis of the difference in attempts between the wild and captive birds, particularly whether differences in number of solves can be explained in differences in number of attempts. For example, do the authors have data on the diversity of solving techniques used by the birds during attempts (e.g. bill vs. foot)? Or were there differences in the number of attempts before solving the first time? I have looked at the data, but I could not find these kinds of details, and our data was restricted to solves and not attempts, which makes looking at the repeatability of attempts impossible. There did appear to be a correlation between success and attempts however, which would be worth looking at more closely. These kinds of analyses might help understand where these differences in performance come from.

4. Multiple times the authors use the phrase “cognitive performance” to refer to performance in a task which supposedly tests cognition. They are not alone in doing this, but I find this phrase very misleading. This task does not measure cognition, it measures behaviour. And variation in behaviour (i.e. performance) can be due to many different factors, of which cognition is only one. I am also not convinced that the phrase is justified by the use of a “cognitive task”, as other authors have suggested, as cognition is used in almost every situation where animals have to acquire or process information or retrieve information from memory. You would struggle to find a behavioural task that did not use cognition in some way! To their credit the authors recognise several of these points in the introduction (lines 88-89), but continue to use the phrase “cognitive performance” throughout the manuscript and clearly consider the task particularly cognitive (e.g. lines 394-6). Considering the variety of factors that can determine performance, I would suggest changing to “performance in a cognitive task” or (given that all tasks could be considered “cognitive”) then “performance in a problem solving task”, “foraging performance” or simply “performance”. This also goes for “learning ability” - which is not the same as performance.

(Sorry for going on about this, “cognitive performance” is a bugbear of mine)

5. I had a few issues with the English in this manuscript and I think the manuscript could do with proof-reading. For example:

- on line 370, it says “However” although the following point seems completely consistent with the one before (no effect of dominance). “However” would be more appropriate at the beginning of line 372.
- on line 322 “long-lived” is an odd choice of words and suggests old rather than what I assume the authors meant - has lived for a long time in captivity.
- on line 386, the wording of the first sentence doesn’t read well, I think it is missing some articles (“the” ecological validity) and pronouns (“our” knowledge).

Minor issues

1. I really liked Table 1 in the appendix and would like to see it in the main manuscript. I understand why the authors have put it in the appendix, particularly if due to word-count issues, but I think it nicely summarises 1. how rare these studies are; and 2. how inconsistent the results are. Maybe a simplified version would work well in the introduction. I would also like to see a bit more discussion (in the intro or discussion) about why there seems to be no consistent picture when comparing wild and captive animals.

2. I was wondering if the repeatability of problem-solving performance could be due to learning? If an individual gets lucky and opens the box, they have an opportunity to learn which could lead to future successes. An individual who never gets any success also never has an opportunity to learn. One way of testing might be to look at time between/attempts required for solving different sections? If individuals are learning, and this leads to apparent differences in ability, then there should be less variation between individuals after having solved the first compartment

(knowledgeable birds solve faster). If individuals do differ, then this should be reflected in each compartment (faster birds solve faster). Again, there was not enough data in the dataset to see this for myself, so I do not know if these data exist (e.g. on video?).

3. Were there individual in total number of attempts/number attempts before solving?
4. Although the authors are completely correct in the penultimate paragraph of their introduction (88-97), it seems a bit out of place in the introduction. Testing across multiple tasks is admirable and sensible, but is not what the authors did here. The different compartments are not really different enough to rule out the other factors influencing performance. I would suggest moving this paragraph to the discussion, and working it into a discussion about the challenges in interpreting how their data relate to variation in cognition.
5. The section starting "The four diverse doors" (lines 128-32) was hard to follow. Consider simplifying this sentence.
6. Did the 3 solves before the box was left open (line 216) have to be consecutive, or could the subject try and open other doors? And did they try to?
7. Lines 230-233 describing excluding birds was a little confusing, and phrased much more clearly in the results. As the reader will encounter this first in the methods, consider making it a bit clearer there as well.

Decision letter (RSOS-181311.R0)

24-Sep-2018

Dear Dr McCune,

The editors assigned to your paper ("The effect of temporary captivity on learning performance in jays") have now received comments from reviewers. We would like you to revise your paper in accordance with the referee and Associate Editor suggestions which can be found below (not including confidential reports to the Editor). Please note this decision does not guarantee eventual acceptance.

Please submit a copy of your revised paper before 17-Oct-2018. Please note that the revision deadline will expire at 00.00am on this date. If we do not hear from you within this time then it will be assumed that the paper has been withdrawn. In exceptional circumstances, extensions may be possible if agreed with the Editorial Office in advance. We do not allow multiple rounds of revision so we urge you to make every effort to fully address all of the comments at this stage. If deemed necessary by the Editors, your manuscript will be sent back to one or more of the original reviewers for assessment. If the original reviewers are not available, we may invite new reviewers.

When submitting your revised manuscript, you must respond to the comments made by the

referees and upload a file "Response to Referees" in "Section 6 - File Upload". Please use this to document how you have responded to the comments, and the adjustments you have made. In order to expedite the processing of the revised manuscript, please be as specific as possible in your response.

- Data accessibility

If you wish to submit your supporting data or code to Dryad (<http://datadryad.org/>), or modify your current submission to dryad, please use the following link:
<http://datadryad.org/submit?journalID=RSOS&manu=RSOS-181311>

- Competing interests

- Authors' contributions

- Acknowledgements

- Funding statement

Please note that Royal Society Open Science charge article processing charges for all new submissions that are accepted for publication. Charges will also apply to papers transferred to Royal Society Open Science from other Royal Society Publishing journals, as well as papers submitted as part of our collaboration with the Royal Society of Chemistry (<http://rsos.royalsocietypublishing.org/chemistry>). If your manuscript is newly submitted and subsequently accepted for publication, you will be asked to pay the article processing charge, unless you request a waiver and this is approved by Royal Society Publishing. You can find out more about the charges at <http://rsos.royalsocietypublishing.org/page/charges>. Should you have any queries, please contact openscience@royalsociety.org.

on behalf of Prof. Kevin Padian (Subject Editor)
openscience@royalsociety.org

Subject Editor's comments:

The reviewers are constructive in their comments on this paper, with good suggestions for revision. Please attend to these carefully; we will ask them to look at your revised version. This is an interesting paper and we appreciate your submission. Good luck.

Comments to Author:

Reviewers' Comments to Author:

Reviewer: 1

Comments to the Author(s)

This is a paper based on a rare dataset on free-ranging individuals being tested on the same cognitive task than wild-caught individuals from the same population. The collection of such synchronized datasets is difficult to achieve, but important if we ought to evaluate the ecological validity of wild vs lab measurements of behaviour or cognition. The topic of the paper is timely and likely to appeal to a large readership.

I have reviewed a previous version of this paper for another journal and found it much improved and clarified. However there are still some issues with statistical analyses, some confusion between concepts, and there is a need to discuss some important limitations.

1-Stats: (a) to deal with unequal number of trials per individual, the authors have opted to repeat all analyses with an additional set of models (3 first trials only). Why not control for 'total number of trials' or 'trial number' as a fixed term in the model instead? The authors included 'Trials' in the analysis of 'time within 2m' and thus should apply this rule consistently. If there is a valid reason not to do so, this rationale should be explained.

(b) Repeatability: trial number must be included as a fixed term here: Biro, P. A., & Stamps, J. A. (2015). Using repeatability to study physiological and behavioural traits: ignore time-related change at your peril. *Animal Behaviour*, 105, 223-230.

2-The authors correctly define the first solution on a given door as problem-solving, and the change in performance in repeated opening of a given door as learning (L252-4). However, the introduction and abstract only concerns 'cognition' in general or learning. The authors should align the text to the data collected and appropriately refer to problem-solving when this is what is referred to or discussed. I make some detailed comments about this below.

Moreover the authors should be consistent and analyze the repeatability of learning speed in addition to that of problem-solving, and discuss the n.s. set of results on learning as well, not just the significant result on problem-solving. Together the picture is a bit more mixed concerning the 3 initial hypotheses than what the authors currently indicate.

3-The very small sample size needs to be part of the discussion; maybe there could be ways to reassure the reader that the significant effect of treatment on problem-solving success is not due to 1-2 individuals and is a general phenomenon?

Detailed comments

L221: "repeatability of learning performance to determine if 221 individual variation in performance represented inherent individual problem solving ability". This is an example where problem-solving is conflated with learning, rephrase to: "repeatability of door-opening success to determine"...

L238: why not control for total nb of trials each jay had, instead of having to conduct several analyses (L230-3), and as done on L243 for motivation? Same for persistence L245.

Definitely need to control for trial number in the survival analysis, as performance is expected to change over trials.

Same and more critical for repeatability (see main comment 1). This is very important as the unequal number of trials could have inflated the number of 0s for birds that had fewer trials, and increased the number of 1s for birds that engaged in several trials. This would reduce within-individual variation and inflate R, potentially explaining such high R=0.86.

L268: I'd give results without interpreting here, cut. Furthermore, the 3rd hypothesis is not fully supported as performance differed only for problem-solving, learning was n.s. Some would say that as cognitive processes are not well understood, only the learning metric is relevant and thus your results rather support the no difference hypothesis.

Where is R for learning? Add to be consistent with problem-solving.

L312-27: this is repetitive of the introduction, I'd keep only 1-2 sentences and instead bring in the main results in this paragraph to highlight them before they get discussed in details.

L329: the wording "simple task" is uninformative and should be avoided throughout. Replace with e.g. extractive foraging task, or problem-solving task.

L335-40: while I appreciate the authors considering this alternative explanation for the n.s. result on time spent within 2m, to me the convincing point would be that they entered this space mainly when the problem was set (i.e. it is not simply their preferred spot, despite having ample other space), and/or they were actively engaged in the task when they did so.

L370: 'however' should be replaced with 'moreover'

L373: you are discussing problem-solving here, not learning.

Motivation as an alternative explanation: Difference in food reward type or possession of cached food in captivity could create differences in motivation? I'd include this point in the discussion on this issue.

L386: before going to concluding paragraph, you need to discuss learning specifically as you only discussed problem-solving: why was there a treatment effect on initial solution speed but not on improvement on the solution? Also, for consistency you should analyze repeatability of learning, which would be discussed here as well.

L438: include R estimate and CIs

L443: the MS is also about learning; what is its relationship to boldness?

Table 1. Many of these studies have NOT measured learning. Adjust caption. Also this portion of the footnote doesn't seem relevant " Previously captive birds with 1-lever experience were no more likely to solve the 2-lever task once released than naïve wild"

Reviewer: 2

Comments to the Author(s)

Overall

I am fully supportive of the aims of this study, and agree with the authors that there is too often an assumption that performance in the laboratory is equivalent to performance in the wild, with only a handful of studies actually testing this. The experiment in this study is well designed, with a few issues I will detail below, and the authors have very sensibly collected data that allows us to understand how differences in motivation can account for differences in performance. I do, however, have a couple of concerns and suggestions, which I hope the authors will consider.

Major concerns

1. There is a major confound to the captivity/wild treatment the authors are testing. The birds in captivity and in the wild were rewarded with different types of food. The authors have good reasons for this decision, but do not discuss the effect this would have on performance. Possible effects include differences in performance or even differences in what is learned. Sulikowski et al. (e.g. Sulikowski, D. & Burke, D. 2010 When a place is not a place: Encoding of spatial information is dependent on reward type. *Behaviour*, 147, 1461-1479), for example, have shown that mynahs seem to learn different types of information or to use different strategies when rewarded with insects versus nectar. I would like to see the authors demonstrate that the differences they observe are due to captivity not differences in food. This could involve comparing the "motivation" of birds with different rewards, although this could be difficult as this is, as far as I can see, perfectly correlated with captivity. If the authors have any data which can tell between these two hypotheses (food vs captivity) I would like to see it included.

2. A second possible confound is competition between birds. In the wild a single bird was allowed access to the apparatus and the other birds were "discouraged" from participating (line 207). In captivity two birds interacted with the task simultaneously. It is difficult to predict a

priori what effect this difference could have - anywhere from social facilitation to competitive exclusion - but I think this deserves some discussion, including why this difference is less important than the difference between captive and wild birds.

3. The authors say that their results cannot be explained by higher motivation for the task in the wild jays (lines 331-2) as there was no difference in the time spent within 2m of the box. There were, however, differences in the number of attempts, with wild jays making more attempts than captive jays. This was discussed in terms of vigilance (quite sensibly in my opinion), but could also reflect differences in motivation or neophobia for the novel apparatus (heightened in unfamiliar surroundings). If that was the case, the differences in solves could be explained in terms of differences in attempts. I would like to see a more detailed analysis of the difference in attempts between the wild and captive birds, particularly whether differences in number of solves can be explained in differences in number of attempts. For example, do the authors have data on the diversity of solving techniques used by the birds during attempts (e.g. bill vs. foot)? Or were there differences in the number of attempts before solving the first time? I have looked at the data, but I could not find these kinds of details, and door data was restricted to solves and not attempts, which makes looking at the repeatability of attempts impossible. There did appear to be a correlation between success and attempts however, which would be worth looking at more closely. These kinds of analyses might help understand where these differences in performance come from.

4. Multiple times the authors use the phrase “cognitive performance” to refer to performance in a task which supposedly tests cognition. They are not alone in doing this, but I find this phrase very misleading. This task does not measure cognition, it measures behaviour. And variation in behaviour (i.e. performance) can be due to many different factor, of which cognition is only one. I am also not convinced that the phrase is justified by the use of a “cognitive task”, as other authors have suggested, as cognition is used in almost every situation where animals have to acquire or process information or retrieve information from memory. You would struggle to find a behavioural task that did not use cognition in some way! To their credit the authors recognise several of these points in the introduction (lines 88-89), but continue to use the phrase “cognitive performance” throughout the manuscript and clearly consider the task particularly cognitive (e.g. lines 394-6). Considering the variety of factors that can determine performance, I would suggest changing to “performance in a cognitive task” or (given that all tasks could be considered “cognitive”) then “performance in a problem solving task”, “foraging performance” or simply “performance”. This also goes for “learning ability” - which is not the same as performance.

(Sorry for going on about this, “cognitive performance” is a bugbear of mine)

5. I had a few issues with the English in this manuscript and I think the manuscript could do with proof-reading. For example:

- on line 370, it says “However” although the following point seems completely consistent with the one before (no effect of dominance). “However” would be more appropriate at the beginning of line 372.
- on line 322 “long-lived” is an odd choice of words and suggests old rather than what I assume the authors meant - has lived for a long time in captivity.
- on line 386, the wording of the first sentence doesn't read well, I think it is missing some articles (“the” ecological validity) and pronouns (“our” knowledge).

Minor issues

1. I really liked Table 1 in the appendix and would like to see it in the main manuscript. I understand why the authors have put it in the appendix, particularly if due to word-count issues, but I think it nicely summarises 1. how rare these studies are; and 2. how inconsistent the results

are. Maybe a simplified version would work well in the introduction. I would also like to see a bit more discussion (in the intro or discussion) about why there seems to be no consistent picture when comparing wild and captive animals.

2. I was wondering if the repeatability of problem-solving performance could be due to learning? If an individual gets lucky and opens the box, they have an opportunity to learn which could lead to future successes. An individual who never gets any success also never has an opportunity to learn. One way of testing might be to look at time between/attempts required for solving different sections? If individuals are learning, and this leads to apparent differences in ability, then there should be less variation between individuals after having solved the first compartment (knowledgeable birds solve faster). If individuals do differ, then this should be reflected in each compartment (faster birds solve faster). Again, there was not enough data in the dataset to see this for myself, so I do not know if these data exist (e.g. on video?).

3. Were there individual in total number of attempts/number attempts before solving?

4. Although the authors are completely correct in the penultimate paragraph of their introduction (88-97), it seems a bit out of place in the introduction. Testing across multiple tasks is admirable and sensible, but is not what the authors did here. The different compartments are not really different enough to rule out the other factors influencing performance. I would suggest moving this paragraph to the discussion, and working it into a discussion about the challenges in interpreting how their data relate to variation in cognition.

5. The section starting "The four diverse doors" (lines 128-32) was hard to follow. Consider simplifying this sentence.

6. Did the 3 solves before the box was left open (line 216) have to be consecutive, or could the subject try and open other doors? And did they try to?

7. Lines 230-233 describing excluding birds was a little confusing, and phrased much more clearly in the results. As the reader will encounter this first in the methods, consider making it a bit clearer there as well.

Author's Response to Decision Letter for (RSOS-181311.R0)

See Appendix A.

RSOS-181311.R1 (Revision)

Review form: Reviewer 1

Is the manuscript scientifically sound in its present form?

Yes

Are the interpretations and conclusions justified by the results?

Yes

Is the language acceptable?

No

Is it clear how to access all supporting data?

Yes

Do you have any ethical concerns with this paper?

No

Have you any concerns about statistical analyses in this paper?

No

Recommendation?

Accept with minor revision (please list in comments)

Comments to the Author(s)

Thank you for these revisions. I still have a few comments to deal with:

- Reply letter: "However, only 7% of the random datasets resulted in a significant treatment effect, a probability of falsely rejecting the null which is very close to the typical alpha criterion of 5%. We have incorporated this information into the discussion in lines: 430-441, 470-487"
Text on line 438 says 5% of randomized datasets; correct the value in the text if it was supposed to be 7%

- Revised L396: "Using several related cognitive tasks and comparing the consistency of an individual's behavior across tasks could inform whether the performance represents an inherent cognitive ability (52)."

This sentence and suggestion is unclear, cut. Instead, if you mean that you do not know which cognitive processes are targeted in your problem-solving task then say this explicitly and refer the reader to papers that have discussed this.

-L456: why "complex" cognitive abilities? This is unrelated to the topic of the paper about ecological relevance. Moreover, these tasks are most probably recruiting simple cognitive abilities (e.g. motor and associative learning), so I'd avoid using 'complex' throughout.

- Revised L522: "problem-solving ability across door types ($R = 0.55 \pm 0.16$, $P = 0.004$)"
But L401: "We found captive and wild jay problem-solving performance across door types was highly repeatable ($R = 0.42 \pm 0.18$, $P = 0.002$)". Contrasting results; adjust.

- Revised L156+175+182: replace "learning" by "problem-solving", as the nature of the task is that of a problem-solving task. You infer learning by using repeated attempts on a given problem-solving task. Please check the MS throughout in link with this.

- Revised L156: something I didn't notice in the first round is that trials seem to have occurred during breeding, is that the case? I'd indicate this explicitly i.e. during breeding or non-breeding season. I assume that there were no ethical concern as the protocol was approved by the Animal Care committee of the university, but this should be clarified.

-Table 1, p.3. Caption: "intended to measure cognition" is not ideal as you cannot confirm that this is what all authors wanted to do, several are innovation tasks that may simply have been interested in innovative problem-solving per se. I suggest replacing by "on several types of problem-solving and cognitive tasks".

Also clarify caption: “Only two studies have statistically compared performance of animals in captivity and in the wild” to “Only two studies have statistically compared performance of wild animals in captivity and in the field”.

I find it very odd to have an entry on Humans; what are “captive humans”?(!) As there is not much space to explain I’d cut this line.

I’m surprised that Orr et al tested pond snails in the field? Do you rather mean that they compared lab-reared lines vs wild-caught animals, but that they were both tested in the lab? If so then this type of comparison is not relevant to the table; it should contain only those studies similar to yours, i.e. tests in the natural vs lab context. Studies comparing animals from a different developmental environment all tested in the lab are irrelevant and should be removed (this may apply to other than Orr et al, e.g. Brust & Guenther?). Please double-check all these references carefully as the reader may not do so and will incorrectly assume that all these studies compared field-based vs lab-based measurement. It is ok if you end up with only a couple of lines in the table; it shows your point that these studies are rare!

Review form: Reviewer 2

Is the manuscript scientifically sound in its present form?

Yes

Are the interpretations and conclusions justified by the results?

Yes

Is the language acceptable?

Yes

Is it clear how to access all supporting data?

Yes

Do you have any ethical concerns with this paper?

No

Have you any concerns about statistical analyses in this paper?

I do not feel qualified to assess the statistics

Recommendation?

Accept as is

Comments to the Author(s)

I am happy with the authors response to my comments, and think the changes they ahve made improve the manuscript.

Decision letter (RSOS-181311.R1)

30-Nov-2018

Dear Dr McCune:

On behalf of the Editors, I am pleased to inform you that your Manuscript RSOS-181311.R1 entitled "Captive jays exhibit reduced problem-solving performance compared to wild conspecifics" has been accepted for publication in Royal Society Open Science subject to minor revision in accordance with the referee suggestions. Please find the referees' comments at the end of this email.

The reviewers and Subject Editor have recommended publication, but also suggest some minor revisions to your manuscript. Therefore, I invite you to respond to the comments and revise your manuscript.

- Ethics statement

- Data accessibility

If you wish to submit your supporting data or code to Dryad (<http://datadryad.org/>), or modify your current submission to dryad, please use the following link:
<http://datadryad.org/submit?journalID=RSOS&manu=RSOS-181311.R1>

- Competing interests

- Authors' contributions

- Acknowledgements

- Funding statement

Because the schedule for publication is very tight, it is a condition of publication that you submit the revised version of your manuscript before 09-Dec-2018. Please note that the revision deadline will expire at 00.00am on this date. If you do not think you will be able to meet this date please let me know immediately.

Supplementary files will be published alongside the paper on the journal website and posted on

the online figshare repository (<https://figshare.com>). The heading and legend provided for each supplementary file during the submission process will be used to create the figshare page, so please ensure these are accurate and informative so that your files can be found in searches. Files on figshare will be made available approximately one week before the accompanying article so that the supplementary material can be attributed a unique DOI.

Please note that Royal Society Open Science charge article processing charges for all new submissions that are accepted for publication. Charges will also apply to papers transferred to Royal Society Open Science from other Royal Society Publishing journals, as well as papers submitted as part of our collaboration with the Royal Society of Chemistry (<http://rsos.royalsocietypublishing.org/chemistry>). If your manuscript is newly submitted and subsequently accepted for publication, you will be asked to pay the article processing charge, unless you request a waiver and this is approved by Royal Society Publishing. You can find out more about the charges at <http://rsos.royalsocietypublishing.org/page/charges>. Should you have any queries, please contact openscience@royalsociety.org.

on behalf of Professor Kevin Padian (Subject Editor)
openscience@royalsociety.org

Associate Editor Comments to Author:

The referees have provided further feedback, and have generally been positive towards your revised work. A few modifications are recommended by one of the reviewers, and the Editors would like to see these incorporated into a final revision before acceptance. Please make sure you tackle these modifications and provide a point-by-point response indicating that these have been made. We look forward to receiving your revision.

Reviewer comments to Author:

Reviewer: 1

Comments to the Author(s)

Thank you for these revisions. I still have a few comments to deal with:

- Reply letter: "However, only 7% of the random datasets resulted in a significant treatment effect, a probability of falsely rejecting the null which is very close to the typical alpha criterion of 5%. We have incorporated this information into the discussion in lines: 430-441, 470-487"
Text on line 438 says 5% of randomized datasets; correct the value in the text if it was supposed to be 7%

- Revised L396: "Using several related cognitive tasks and comparing the consistency of an individual's behavior across tasks could inform whether the performance represents an inherent cognitive ability (52)."

This sentence and suggestion is unclear, cut. Instead, if you mean that you do not know which cognitive processes are targeted in your problem-solving task then say this explicitly and refer the reader to papers that have discussed this.

-L456: why “complex” cognitive abilities? This is unrelated to the topic of the paper about ecological relevance. Moreover, these tasks are most probably recruiting simple cognitive abilities (e.g. motor and associative learning), so I’d avoid using ‘complex’ throughout.

- Revised L522: “problem-solving ability across door types ($R = 0.55 \pm 0.16$, $P = 0.004$)”
But L401: “We found captive and wild jay problem-solving performance across door types was highly repeatable ($R = 0.42 \pm 0.18$, $P = 0.002$)”. Contrasting results; adjust.

- Revised L156+175+182: replace “learning” by “problem-solving”, as the nature of the task is that of a problem-solving task. You infer learning by using repeated attempts on a given problem-solving task. Please check the MS throughout in link with this.

- Revised L156: something I didn’t notice in the first round is that trials seem to have occurred during breeding, is that the case? I’d indicate this explicitly i.e. during breeding or non-breeding season. I assume that there were no ethical concern as the protocol was approved by the Animal Care committee of the university, but this should be clarified.

-Table 1, p.3. Caption: “intended to measure cognition” is not ideal as you cannot confirm that this is what all authors wanted to do, several are innovation tasks that may simply have been interested in innovative problem-solving per se. I suggest replacing by “on several types of problem-solving and cognitive tasks”.

Also clarify caption: “Only two studies have statistically compared performance of animals in captivity and in the wild” to “Only two studies have statistically compared performance of wild animals in captivity and in the field”.

I find it very odd to have an entry on Humans; what are “captive humans”?(!) As there is not much space to explain I’d cut this line.

I’m surprised that Orr et al tested pond snails in the field? Do you rather mean that they compared lab-reared lines vs wild-caught animals, but that they were both tested in the lab? If so then this type of comparison is not relevant to the table; it should contain only those studies similar to yours, i.e. tests in the natural vs lab context. Studies comparing animals from a different developmental environment all tested in the lab are irrelevant and should be removed (this may apply to other than Orr et al, e.g. Brust & Guenther?). Please double-check all these references carefully as the reader may not do so and will incorrectly assume that all these studies compared field-based vs lab-based measurement. It is ok if you end up with only a couple of lines in the table; it shows your point that these studies are rare!

Reviewer: 2

Comments to the Author(s)

I am happy with the authors response to my comments, and think the changes they ahve made improve the manuscript.

Author's Response to Decision Letter for (RSOS-181311.R1)

See Appendix B.

Decision letter (RSOS-181311.R2)

12-Dec-2018

Dear Dr McCune,

I am pleased to inform you that your manuscript entitled "Captive jays exhibit reduced problem-solving performance compared to wild conspecifics" is now accepted for publication in Royal Society Open Science.

on behalf of Professor Kevin Padian (Subject Editor)
openscience@royalsociety.org

Appendix A

Dear Reviewers,

We appreciate the constructive comments and we have addressed each below. We have revised our manuscript accordingly, and these edits are color-coded in the manuscript such that each reviewer can clearly see where we have made the changes that they requested. The changes in response to comments from Reviewer 1 are in **Green**, the changes recommended by Reviewer 2 are in **Purple**. The changes that both reviewers wanted to see are indicated in **Red**.

Responses to Reviewer 1's comments:

We are glad you found our manuscript improved! We agree that the last round of comments you gave us were very helpful and pointed out where increased clarity was needed.

1. Stats: (a) to deal with unequal number of trials per individual, the authors have opted to repeat all analyses with an additional set of models (3 first trials only). Why not control for 'total number of trials' or 'trial number' as a fixed term in the model instead? The authors included 'Trials' in the analysis of 'time within 2m' and thus should apply this rule consistently. If there is a valid reason not to do so, this rationale should be explained. Related detailed comment: L238: why not control for total nb of trials each jay had, instead of having to conduct several analyses (l230-3), and as done on l243 for motivation? Same for persistence l245.

Author response: We think of these as two separate problems: Accounting for opportunity to solve by including the amount of time jays *chose* to attend to the task, and accounting for a potential confound where several jays in the captive treatment were not *allowed* equal opportunity to interact with the task.

We included the time within 2m offset in our quasipoisson models for the dependent variables "successes" and "attempts" to account for differences in opportunity to solve based on whether jays were choosing to attend to the task within trials. Trial number is not as good of an indicator of the opportunity that jays had to interact with the puzzle box because many distractions (social interaction, other foraging, preening, etc) lead to jays not spending the whole trial within 2m attending to the task. As a result, the maximum number of

trials that a subject experienced was not correlated with the summed amount of time they spent within 2m during trials ($r = 0.23$).

We agree that it was confusing to then include Trial in the analysis examining the effect of treatment on time spent within 2m. Originally this was to account for the opportunity jays had to be within 2m because they were given a trial. However, since we occasionally ended trials early due to jays completing all solves, or a lack of participation, a more detailed way to control for opportunity to be within 2m is to instead include the summed total length of trial time (in seconds) for each subject as a model offset. This allows us to determine the proportion of time during trials that jays chose to spend attending to the task.

The additional models with the subset data were important to ensure that our captive sample was not biased by these individuals that were not given as many opportunities to solve, a situation that did not occur in our wild treatment condition. Since trial times for subjects were different, it made the most sense to subset the whole dataset based on trial number rather than summed trial times.

We have clarified the methods accordingly in lines: 215-217, 232-240, 245-248

We have also edited the results from our Time within 2m model in Table 1. When the summed total trial time offset is included, we no longer see a significant effect of dominance rank.

1. (b) Repeatability: trial number must be included as a fixed term here: Biro, P. A., & Stamps, J. A. (2015). Using repeatability to study physiological and behavioural traits: ignore time-related change at your peril. *Animal Behaviour*, 105, 223-230. Related detailed comment: Same and more critical for repeatability (see main comment 1). This is very important as the unequal number of trials could have inflated the number of 0s for birds that had fewer trials, and increased the number of 1s for birds that engaged in several trials. This would reduce within-individual variation and inflate R, potentially explaining such high $R=0.86$.

Author response: Thank you for directing us to that relevant and helpful paper, we had not previously seen it. We have incorporated your suggestion and made changes in lines: 515-520

2. (a) The authors correctly define the first solution on a given door as problem-solving, and the change in performance in repeated opening of a given door as learning (L252-4). However, the introduction and abstract only concerns 'cognition' in general or learning. The authors should align the text to the data collected and appropriately refer to problem-solving when this is what is referred to or discussed. I make some detailed comments about this below.

Author response: Thank you for pointing out this gap! We have made edits accordingly in lines: 11-13, 399

2. (b) Moreover the authors should be consistent and analyze the repeatability of learning speed in addition to that of problem-solving, and discuss the n.s. set of results on learning as well, not just the significant result on problem-solving. Together the picture is a bit more mixed concerning the 3 initial hypotheses than what the authors currently indicate.
Related detailed comments: Where is R for learning? Add to be consistent with problem-solving. AND L386: before going to concluding paragraph, you need to discuss learning specifically as you only discussed problem-solving: why was there a treatment effect on initial solution speed but not on improvement on the solution? Also, for consistency you should analyze repeatability of learning, which would be discussed here as well.

Author response: We have added this important information that we overlooked previously. These additional analyses are described in the appendix in lines: 515-523

And we discuss the non-significant results of learning performance and repeatability in lines: 446-447, 449-455

3. The very small sample size needs to be part of the discussion; maybe there could be ways to reassure the reader that the significant effect of treatment on problem-solving success is not due to 1-2 individuals and is a general phenomenon?

Author response: This is a valid concern. To check that, with this sample size, there was not a chance we could come by these results at random we

simulated 1000 datasets identical to our original data except that treatment condition assignment was shuffled. In these datasets the bird ID and solving performance are randomly decoupled from the treatment condition. We then ran the same quasipoisson model, analyzing total number of successes as a function of treatment, dominance rank and including the time within 2m offset. We then determined the proportion of datasets in which we found a significant effect of treatment. If our observed results are attributable to a few exceptional individuals, then in 1000 randomized datasets we would expect a high probability those individuals would end up together in one treatment group and we would see a significant treatment effect. However, only 7% of the random datasets resulted in a significant treatment effect, a probability of falsely rejecting the null which is very close to the typical alpha criterion of 5%. We have incorporated this information into the discussion in lines: 430-441, 470-487

L221: “repeatability of learning performance to determine if 221 individual variation in performance represented inherent individual problem solving ability”. This is an example where problem-solving is conflated with learning, rephrase to: “repeatability of door-opening success to determine”...

Author response: Good point, we have made this suggested change in lines: 399

Definitely need to control for trial number in the survival analysis, as performance is expected to change over trials.

Author response: Change over time is explicitly modeled in survival analysis, where the dependent variable is the time until the event occurs (in our paper, either a first solve on any door, or time between a first and last solve of a specific door). Again, instead of trial number here we used trial time, summed across trials (where relevant because some jays had a first and last solve within one trial). We apologize, because we now see that this was not clear in our description of this analytical technique in the methods so we have clarified this in lines: 257-265

L268: I'd give results without interpreting here, cut. Furthermore, the 3rd hypothesis is not fully supported as performance differed only for problem-solving, learning was n.s. Some would say that as cognitive processes are not well understood, only the learning metric is relevant and thus your results rather support the no difference hypothesis.

Author response: We *a priori* did not expect a difference between problem-solving performance and learning speed, therefore our hypotheses referred to the two combined. Additionally, our study was not explicitly designed to elucidate detailed cognitive mechanisms explaining performance on our experimental task. Rather, we hoped to demonstrate that regardless of the cognitive trait, the *behavior* of individuals towards an experimental assessment can be different in captivity than for wild subjects. This has largely not been considered in the comparative cognition field, and confounds our conclusions about what cognitive mechanisms may be present in different species. However, in light of our other changes disentangling problem-solving and learning ability, we have clarified our hypotheses to represent both.

We have made the suggested edit to the results, and changed the wording of the hypotheses in lines: 79-85

L312-27: this is repetitive of the introduction, I'd keep only 1-2 sentences and instead bring in the main results in this paragraph to highlight them before they get discussed in details.

Author response: We have made these changes in lines: 315-321

L329: the wording "simple task" is uninformative and should be avoided throughout. Replace with e.g. extractive foraging task, or problem-solving task.

Author response: We have made these changes throughout (e.g. line 10)

L335-40: while I appreciate the authors considering this alternative explanation for the n.s. result on time spent within 2m, to me the convincing point would be that they entered this space mainly when the problem was set (i.e. it is not simply their preferred spot, despite having ample other space), and/or they were actively engaged in the task when they did so.

Author response: Jays preferred the far end of the aviary from the table where we put the puzzle box because they knew that experimenters reached through the window to access the table for adding food or manipulating the puzzle box doors. Additionally, the door into the aviaries was on the same wall as the window. Knowing this, we set up the aviaries with extensive foliage in the

upper corner furthest from the window and doors so the jays had somewhere to feel safe. We have clarified this in lines: 349-351

L370: 'however' should be replaced with 'moreover'

Author response: We ended up deleting this sentence.

L373: you are discussing problem-solving here, not learning.

Author response: We have removed this paragraph because dominance is no longer a significant predictor of performance.

Motivation as an alternative explanation: Difference in food reward type or possession of cached food in captivity could create differences in motivation? I'd include this point in the discussion on this issue.

Author response: We have added additional analyses of other variables relating to the interaction behaviors of jays at the puzzle box to help substantiate that the differences in food reward do not affect the interaction of jays at the puzzle box. Jays in our wild treatment also had access to cached food from the previous season as well as the peanuts that we gave them in the puzzle box during habituation and trials. We've also elaborated on the different food rewards as a confound in the discussion. Changes in lines: 193-200, 342-345, 354-369

L438: include R estimate and CIs

Author response: Good point, we apologize for not including this before! We have made this edit in line: 494

L443: the MS is also about learning; what is its relationship to boldness?

Author response: We added this analysis and found that learning speed is also unrelated to boldness. Changes made in lines: 449-450, 500-505

Table 1. Many of these studies have NOT measured learning. Adjust caption. Also this portion of the footnote doesn't seem relevant " Previously captive birds with 1-lever experience were no more likely to solve the 2-lever task once released than naïve wild'

Author response: We have moved this table into the body of the paper and made these changes in the caption of the table on page 3

Responses to Reviewer 2's comments:

1. There is a major confound to the captivity/wild treatment the authors are testing. The birds in captivity and in the wild were rewarded with different types of food. The authors have good reasons for this decision, but do not discuss the effect this would have on performance. Possible effects include differences in performance or even differences in what is learned. Sulikowski et al. (e.g. Sulikowski, D. & Burke, D. 2010 When a place is not a place: Encoding of spatial information is dependent on reward type. *Behaviour*, 147, 1461-1479), for example, have shown that mynahs seem to learn different types of information or to use different strategies when rewarded with insects versus nectar. I would like to see the authors demonstrate that the differences they observe are due to captivity not differences in food. This could involve comparing the "motivation" of birds with different rewards, although this could be difficult as this is, as far as I can see, perfectly correlated with captivity. If the authors have any data which can tell between these two hypotheses (food vs captivity) I would like to see it included.

Author response: We have considered the potential impact of different food rewards for each treatment, and discussed these much more thoroughly in the text. To address this concern, and the similar concern voiced in your comment #3, we have incorporated 3 additional variables describing behavioral interactions with the puzzle box: contacts - touches to the puzzle box not on the doors, lands - sitting on top of the puzzle box, visits - number of separate instances that the subject comes within 2m of the puzzle box. Originally, we did not think it was necessary to include these as they are behaviors that do not immediately relate to solving the puzzle box doors. However, now we see that they may help to clarify whether jays in each treatment differ in motivation.

Additionally, we have clarified that we did originally try to use peanuts, but captive jays would not engage at the same level as wild jays. When we switched to using worms, a food item that we did not give them in their

maintenance diet, the interactions drastically increased. Furthermore, captive subjects were food deprived although they could recover caches around the aviary.

It is possible that the different reward types could lead to different foraging strategies and performance at the task that is not comparable. However, we believe that mobile insect prey would lead captive jays to adopt a much more rapid, attack foraging behavior compared to that needed to forage for mast.

In all of these cases, the potential predicted change would lead the captive jays to outperform the wild jay subjects. Therefore, we believe the use of different food reward types resulted in a more conservative comparison that minimized the likelihood of finding reduced performance in captive jays.

Changes in lines: 342-345, 354-368

2.A second possible confound is competition between birds. In the wild a single bird was allowed access to the apparatus and the other birds were “discouraged” from participating (line 207). In captivity two birds interacted with the task simultaneously. It is difficult to predict a priori what effect this difference could have - anywhere from social facilitation to competitive exclusion - but I think this deserves some discussion, including why this difference is less important than the difference between captive and wild birds.

Author response: To address the concern about social competition we modeled the effect of dominance rank on our interaction variables describing interactions at the puzzle box: Successes, attempts, contacts, lands, and visits for *only* the captive jays. Dominance did not have a significant impact on any of these variables (Successes $\beta = 0.25$, $p = 0.23$; Attempts $\beta = 0.14$, $p = 0.32$; Contacts $\beta = -0.06$, $p = 0.79$; Lands $\beta = 0.19$, $p = 0.15$; Visits $\beta = 0.06$, $p = 0.55$), indicating that dominance relationships within the dyad of jays in captivity did not change *who* interacted at our task. We will also note that in the wild, jays more dominant than the focal subject could come in to displace the subject at the puzzle box before we could prevent it. However, we don't remember this ever happening.

Secondly, based on our unpublished social learning experiment conducted with this population, we do think it is likely that social facilitation could have a role in increased engagement with the task. However, we think this effect is likely to be similar in the captive and wild treatments because the non-subject wild jays in each group were allowed to eat from the open puzzle box prior to

trials (before any trials began, as well as before each individual trial started). Only after we closed the puzzle box doors and began the trial did we discourage non-subject jays from interacting with the task. Frequently, several group members were within 10m of the task at the same time as the focal individual, resulting in a similar social environment as that experienced by captive subjects.

We have made changes clarifying these details in lines: 105-107, 174-177, 287-292, 414-417

3. The authors say that their results cannot be explained by higher motivation for the task in the wild jays (lines 331-2) as there was no difference in the time spent within 2m of the box. There were, however, differences in the number of attempts, with wild jays making more attempts than captive jays. This was discussed in terms of vigilance (quite sensibly in my opinion), but could also reflect differences in motivation or neophobia for the novel apparatus (heightened in unfamiliar surroundings). If that was the case, the differences in solves could be explained in terms of differences in attempts. I would like to see a more detailed analysis of the difference in attempts between the wild and captive birds, particularly whether differences in number of solves can be explained in differences in number of attempts. For example, do the authors have data on the diversity of solving techniques used by the birds during attempts (e.g. bill vs. foot)? Or were there differences in the number of attempts before solving the first time? I have looked at the data, but I could not find these kinds of details, and door data was restricted to solves and not attempts, which makes looking at the repeatability of attempts impossible. There did appear to be a correlation between success and attempts however, which would be worth looking at more closely. These kinds of analyses might help understand where these differences in performance come from.

Author response: These jays only used their beak to interact with the puzzle box. Unfortunately, we did not collect the data on attempts at that fine of a scale that we could determine the number of attempts before each solve. However, to help compare motivation of jays in each treatment we have added additional data on other interaction behaviors (as mentioned in response to your comment #1) that are not directly functional for opening puzzle box doors. Because jay subjects do not differ in these other, non-functional behaviors, we believe the motivation of jays to engage with the task was equal, but the ability to problem-solve was disrupted by neophobia in the new environment.

We did look at repeatability of attempts across trials (see response to your minor comment #3), but you are right that the more interesting question would be repeatability of attempts before each solve. We hope future experiments can answer this question!

Changes in lines: 193-200, 342-345, Table 2, Table A1

4. Multiple times the authors use the phrase “cognitive performance” to refer to performance in a task which supposedly tests cognition. They are not alone in doing this, but I find this phrase very misleading. This task does not measure cognition, it measures behaviour. And variation in behaviour (i.e. performance) can be due to many different factors, of which cognition is only one. I am also not convinced that the phrase is justified by the use of a “cognitive task”, as other authors have suggested, as cognition is used in almost every situation where animals have to acquire or process information or retrieve information from memory. You would struggle to find a behavioural task that did not use cognition in some way! To their credit the authors recognise several of these points in the introduction (lines 88-89), but continue to use the phrase “cognitive performance” throughout the manuscript and clearly consider the task particularly cognitive (e.g. lines 394-6). Considering the variety of factors that can determine performance, I would suggest changing to “performance in a cognitive task” or (given that all tasks could be considered “cognitive”) then “performance in a problem solving task”, “foraging performance” or simply “performance”. This also goes for “learning ability” - which is not the same as performance.

(Sorry for going on about this, “cognitive performance” is a bugbear of mine)

Author response: We understand this pet peeve, it does make sense. We have made these changes throughout (e.g. 315-316)

5. I had a few issues with the English in this manuscript and I think the manuscript could do with proof-reading. For example:
- on line 370, it says “However” although the following point seems completely consistent with the one before (no effect of dominance). “However” would be more appropriate at the beginning of line 372.

- on line 322 “long-lived” is an odd choice of words and suggests old rather than what I assume the authors meant - has lived for a long time in captivity.

-on line 386, the wording of the first sentence doesn't read well, I think it is missing some articles (“the” ecological validity) and pronouns (“our” knowledge).

Author response: We apologize for the lack of clarity. We have made changes in line with this comment in lines: 456-458

Minor issues

1.I really liked Table 1 in the appendix and would like to see it in the main manuscript. I understand why the authors have put it in the appendix, particularly if due to word-count issues, but I think it nicely summarises 1. how rare these studies are; and 2. how inconsistent the results are. Maybe a simplified version would work well in the introduction. I would also like to see a bit more discussion (in the intro or discussion) about why there seems to be no consistent picture when comparing wild and captive animals.

Author response: We are glad you like this table! We have moved it to the main body of the paper on page 3. We have added more discussion about the inconsistency of performance of captive and wild subjects in the discussion in lines: 324-339

2.I was wondering if the repeatability of problem-solving performance could be due to learning? If an individual gets lucky and opens the box, they have an opportunity to learn which could lead to future successes. An individual who never gets any success also never has an opportunity to learn. One way of testing might be to look at time between/attempts required for solving different sections? If individuals are learning, and this leads to apparent differences in ability, then there should be less variation between individuals after having solved the first compartment (knowledgeable birds solve faster). If individuals do differ, then this should be reflected in each compartment (faster birds solve faster). Again, there was not enough data in the dataset to see this for myself, so I do not know if these data exist (e.g. on video?).

Author response: This is an interesting point! If, after opening a given door type 3 times, jays generally learn something about how to successfully interact with the puzzle box doors to open them, then we would expect a decrease in the latency between successive solves on subsequent door types. We modeled time between solves on different door types (time between the last opening of one door type and the first opening of the next door type) as a function of treatment and solve number (second, third or fourth door type solved). There was a negative trend, especially for captive subjects, suggesting there is some learning or improvement in problem-solving ability across the experiment (see Figure 1 below). However, this trend was not significant for solve number. We did find a significant treatment effect with supports the results from our cox-proportional hazard model of problem-solving performance. We incorporated this information in lines: 403-406, 537-553, Figure A2

Figure 1: Boxplot showing the time between solves on different door types (i.e. data for the “Second” door solved shows the latency from the last opening of the first door type to the first opening of the second door type, etc.).

3. Were there individual [differences] in total number of attempts/number attempts before solving?

Author response: In our data we only have a total number of attempts per trial, and unfortunately cannot distinguish the number of attempts before any particular solve. But we tested whether individuals consistently differ in the

number of attempts in each trial. We first tried to use the rptR package for Poisson distributed data as we had with our previous repeatability models. We included a random effect of bird ID, and included treatment as a fixed effect. We excluded the 3 jays that only had 1 trial because they solved all doors in that time. This model would not converge, so we then assessed the amount of deviance explained by the ID random effect with a likelihood ratio test. We found that inclusion of this random effect significantly improved the fit of the model ($\chi^2 = 800.7, P < 0.001$), indicating that number of attempts per time across trials were clustered within bird. We think these results merit further investigation with more detailed data on number of attempts before a solve on each door. With our more broad-scale data on attempts we cannot tell how a solve (or solves) during a trial affected the attempt number. We added this information in lines: 523-535

4. Although the authors are completely correct in the penultimate paragraph of their introduction (88-97), it seems a bit out of place in the introduction. Testing across multiple tasks is admirable and sensible, but is not what the authors did here. The different compartments are not really different enough to rule out the other factors influencing performance. I would suggest moving this paragraph to the discussion, and working it into a discussion about the challenges in interpreting how their data relate to variation in cognition.

Author response: We understand the hesitation to identify the different door types on our puzzle box as distinct problem-solving tasks. As suggested, we have moved the details about repeatability of performance at the puzzle box to the discussion, and the methods to the appendix. Changes to lines: 393-400, 507-535

5. The section starting “The four diverse doors” (lines 128-32) was hard to follow. Consider simplifying this sentence.

Author response: Thank you for pointing this out, we hope we have clarified it accordingly in lines: 103-105

6. Did the 3 solves before the box was left open (line 216) have to be consecutive, or could the subject try and open other doors? And did they try to?

Author response: No, the 3 solves did not have to be consecutive. 6 birds switched door types after solving a door a first or second time. This door switching was more prevalent in wild (4 jays) than captive subjects (2 jays). We clarified this in the methods in lines: 201-203

7.Lines 230-233 describing excluding birds was a little confusing, and phrased much more clearly in the results. As the reader will encounter this first in the methods, consider making it a bit clearer there as well.

Author response: Thank you for your feedback here! We have rephrased this piece to make it clearer in lines: 215-221

Appendix B

Dear Reviewers,

Thank you again for your original comments that have much improved our manuscript. We have now made the additional edits that you requested. The changes in response to comments from Reviewer 1 are in **Green** throughout the revised manuscript.

Responses to Reviewer 1's comments:

We really appreciate your thorough read-through to catch inconsistencies arising during the revision process. We have addressed all of your comments below.

1. We changed our statements regarding the proportion of random datasets resulting in significant treatment effects from 5% to 7% in lines: 442 – 445 and 486 – 489.
2. We have clarified and cut down the sentence about cross-task consistency in problem-solving performance in lines: 400 – 403.
3. You are right that our task does not require more complex cognitive abilities than associative learning. We have removed this terminology in lines 323, 460 and throughout the manuscript.
4. Apologies for the inconsistency in the repeatability results that we reported. We ran the repeatability model on binarized door-opening success data and used a logistic repeatability model (as described in the methods). However, we also tried analyzing the full count data with a Poisson repeatability model to ensure we would get the same result. We mistakenly included the results from the Poisson model in the appendix but we have fixed that in line 526.
5. Thank you for pointing out the oversight of saying “learning trials” when this is not appropriate. We have changed this in lines 157, 183, and removed that terminology elsewhere in the manuscript.
6. We observed the breeding behaviors of all flocks. We only took jays into captivity once the young fledged and were seen foraging independently, and we did not observe further successful nests. As resident species with a range in the southern U.S., Mexican Jays breed early and there were already fledged, independent young by mid-May. We have clarified in this in lines: 112 – 114.
7. We have made the changes that you requested to Table 1 on page 4, and agree that it is now a much simpler table for the reader to comprehend.